

# Leveraging a Disdrometer Network to Develop a Probabilistic Precipitation Phase Model in Eastern Canada

Alexis Bédard-Therrien[1], François Anctil[1], Julie M. Thériault[2], Olivier Chalifour[2], Fanny Payette[3], Alexandre Vidal[3] and Daniel F. Nadeau[1]

[1]Département de génie civil et de génie des eaux, Université Laval, Québec, QC, Canada
[2]Département des sciences de la Terre et de l'atmosphère, Université du Québec à Montréal, Montréal, QC, Canada
[3]Hydro-Québec, Direction Planification de la conduite du système énergétique, Montréal, QC, Canada

Correspondence to: alexis.bedard-therrien.1@ulaval.ca

**Abstract**. This study presents a probabilistic model that partitions the precipitation phase based on hourly measurements from a network of radar-based disdrometers in eastern Canada. The network consists of 27 meteorological stations located in a boreal climate for the years 2020-2023. Precipitation phase observations showed a 2-m air temperature interval between 0-4°C where probabilities of occurrence of solid, liquid, or mixed precipitation significantly overlapped. Single-phase precipitation was also found to occur more frequently than mixed-phase precipitation. Probabilistic phase-guided partitioning (PGP) models of increasing complexity using random forest algorithms were developed. The PGP models classified the precipitation phase and partitioned the precipitation accordingly into solid and liquid amounts. PGP_basic is based on 2-m air temperature and site elevation, while PGP_hydromet integrates relative humidity. PGP_full includes all the above data plus atmospheric reanalysis data. The PGP models were compared to benchmark precipitation phase partitioning methods. These included a single temperature threshold model set at 1.5°C, a linear transition model with dual temperature thresholds of –0.38 and 5°C, and a psychrometric balance model. Among the benchmark models, the single temperature threshold had the best classification performance (F1 score of 0.74) due to a low count of mixed-phase events. The other benchmark models tended to over-predict mixed-phase precipitation in order to decrease partitioning error. All PGP models showed significant phase classification improvement by reproducing the observed overlapping precipitation phases based on 2-m air temperature. PGP_hydromet and PGP_full displayed the best classification performance (F1 score of 0.84). In terms of partitioning error, PGP_full had the lowest RMSE (0.27 mm) and the least variability in performance. The RMSE of the single temperature threshold model was the highest (0.40 mm) and showed the greatest performance variability. An input variable importance analysis revealed that the additional data used in the more complex PGP models mainly improved mixed-phase precipitation prediction. The improvement of mixed-phase prediction remains a challenge. Relative humidity was deemed the least important input variable used, due to consistent near water vapor saturation conditions.





Additionally, the reanalysis atmospheric data proved to be an important factor to increase the robustness of the partitioning process. This study establishes a basis for integrating automated phase observations into a hydrometeorological observation network and developing probabilistic
precipitation phase models.

## 1   Introduction

Precipitation phase is a critical component in hydrological modelling. Simply put, the hydrological effect following either a snowfall or rainfall event are drastically different; snowfall accumulates in winter and melts later in the spring or during winter melt events, while rain can
either infiltrate or become runoff, potentially increasing streamflow in the short term. The precipitation phase also affects snowpack characteristics in different ways; a rain-on-snow event infiltrates the snowpack, increasing its liquid water content, and can create ice layers and change the snowpack internal characteristics (Singh et al., 1997; Wever et al., 2016), while snowfall increases the depth of the snowpack. During individual precipitation events, errors in snowpack
water equivalent (SWE) and depth are mainly caused by errors in precipitation partitioning (Leroux et al., 2023). On a seasonal basis, the precipitation phase significantly affects the ablation of the snowpack, particularly due to its impact on the snow albedo (Essery et al., 2013; Günther et al., 2019). As such, the cumulative effects of misclassified precipitation have a significant impact on various seasonal values such as peak SWE, peak discharge date, and snow
cover duration  (Harder and Pomeroy, 2014).

As the climate warms, regions that typically experience winter snowfall are expected to face more rainfall in their winter precipitation, resulting in more rain-on-snow events (Jeong and Sushama, 2018; Ye et al., 2008; Musselman et al., 2018). Consequently, the proportion of runoff caused by rain-on-snow events during the winter is projected to increase in these regions (Jeong
and Sushama, 2018; Musselman et al., 2018). Effective precipitation partitioning methods are more important than ever to anticipate potentially damaging events and to monitor water resources at the catchment scale. This need is also felt for field monitoring of precipitation quantities. Indeed, solid precipitation is much more sensitive to undercatch (underestimation due to the wind moving hydrometeors away from the gauge) than liquid precipitation (Rasmussen et
al., 2012). Consequently, an inaccurate measurement of the phase necessarily translates into an erroneous estimation of the precipitation quantity.

The modelling of the precipitation phase in operational hydrological models is often based on a single near-surface air temperature threshold (Harpold et al., 2017). While simple to implement, this method cannot predict mixed precipitation events, which tend to occur when falling
hydrometeors of different sizes coexist and melt at different rates (Thériault and Stewart, 2010). As an alternative to the single-threshold approach, one can use a linear relationship to account





for mixed-phase precipitation events while occurring between those two thresholds. Furthermore, these type of methods can be refined into curvilinear functions, which would theoretically yield to a more accurate phase identification, but using more computation resources (Feiccabrino et al., 2013). In both cases, the classification error is reduced when compared to the single threshold approach (Feiccabrino et al., 2013; Wen et al., 2013). The advantage of using temperature threshold-based models comes mainly from a data availability and computational requirement standpoint. Variables other than air temperature are known to influence the precipitation phase, such as relative humidity and atmospheric pressure (Behrangi et al., 2018; Dai, 2008; Jennings et al., 2018). Thus, precipitation partitioning models can be improved by using dew point temperature (e.g. Marks et al., 2013; Ye et al., 2013) or wet bulb temperature (e.g. Ding et al., 2014; Wang et al., 2019; Behrangi et al., 2018) instead of relying solely on air temperature, increasing the spatial robustness of such models.

Phase partitioning models tend to rely on near-surface hydrometeorological variables because this information is easily accessible. However, the hydrometeor's initial phase as it leaves the cloud, the shape and size distribution of the droplets, and the properties of the atmosphere from the cloud to the ground all determine the precipitation phase (Feiccabrino et al., 2015). As it falls, the hydrometeor exchanges latent and sensible heat with its surroundings, linking its phase to the temperature and vapour deficit as they fall through the atmosphere. Additionally, both heat fluxes are also affected by the ventilation of the hydrometeor, which depends on its fall speed and the surrounding wind velocities (Stewart, 1992).

Atmospheric temperature gradients can vary with time, so the thickness of the melting atmospheric layer is also a key variable to consider, as it affects the time the hydrometeor spends in conditions favourable to melting. Empirical models approximate this layer thickness by computing the height difference between two selected pressure levels (Feiccabrino et al., 2015). Precipitation rates can also increase the energy required to melt hydrometeors. Indeed, at high precipitation rates, there is a larger volume to melt, thus increasing the likelihood of solid precipitation at warmer temperatures (Froidurot et al., 2014; Thériault and Stewart, 2010). Therefore, by accounting for the characteristics of the atmospheric layer, microphysical models can determine the precipitation phase of falling hydrometeors (Thériault and Stewart, 2010). Other models are instead based on the statistical relationship between the hydrometeorological variables and the precipitation phase. Such models compute the probability of a precipitation phase occurring considering a set of environmental conditions.

The methodology for calculating the probability of phase occurrence varies across studies and includes, for example, a curvilinear function (Dai, 2008), logistic regression (Behrangi et al., 2018; Froidurot et al., 2014; Jennings et al., 2018), and machine learning algorithms (Shin et al., 2022). Therefore, these methods output a precipitation type rather than a fraction of solid and



liquid precipitation, in the case of dual threshold models. However, the use of these methods is limited when dealing with mixed-phase precipitation, as they do not provide information on how

the precipitation is partitioned. Fortunately, mixed-phase events are less common than single-phase events (Dai, 2008) and are thus often omitted from studies using probabilistic methods.

In addition to selecting the appropriate variables to include in a phase partitioning model, the quality and availability of the validation dataset is a critical aspect to consider. Indeed, the scarcity of validation data was cited by Harpold et al. (2017) as a major factor hindering the

development of phase partitioning models. Direct manual phase observations collected from trained observers have been used to validate precipitation partitioning models (e.g. Behrangi et al., 2018; Dai, 2008; Froidurot et al., 2014; Jennings et al., 2018). Jennings et al. (2023) have also shown the possibility of using crowd-sourced precipitation phase data. While such datasets are extensive in time and space, they do not provide quantitative information on the snow and

rain fractions in mixed-phase events, thus limiting the possible predicted precipitation phases to either solid or liquid.

High-frequency automatic measurements do not suffer from limitations caused by mixed-phase precipitation (Froidurot et al., 2014; Harpold et al., 2017). One possible validation approach based on automatic data is to use precipitation measurements collocated with snow cover height

measurements (Harder and Pomeroy, 2013; Marks et al., 2013). A more direct automatic approach is to use a disdrometer, which identifies the phase of the hydrometeor according to its size and falling speed. For instance, Wayand et al. (2016) utilized a disdrometer to associate precipitation phase with precipitation amounts, which helped evaluate multiple phase models. This combination of observations not only allows for the validation of a phase model, but also

addresses a major limitation of previous studies, namely the partitioning of precipitation in the case of mixed-phase events. Another important factor to consider is the time step of the validation data. While many conceptual hydrological models employ daily timesteps to determine precipitation phase, sub-daily time steps greatly enhance the accuracy of modelled phase (Feiccabrino, 2020; Harder and Pomeroy, 2013). Therefore, it is necessary to use sub-daily

time steps, such as 15 minutes or hourly, as a significant portion of the phase model's performance depends on the time step of interest.

There are many ways to improve the representation of the precipitation phase for hydrological purposes. As pointed out in Harpold et al. (2017), the often too simple phase models need more hydrometeorological observations for a successful partitioning of the precipitation phase. Such

observations include the relative humidity, as well as atmospheric information, like the temperature and humidity lapse rate. Additionally, an important research limitation comes from the lack of validation data. Direct observations, while commonly utilized, have limited application for mixed phase precipitation due to their qualitative nature. A preferable solution





involves automated phase observations, as they enable the coupling with precipitation rate
measurements. Additionally, direct phase observations datasets indicate the existence of a
temperature transition zone, where both snow and rain are possible. This highlights the
limitations of simplistic phase models that fail to capture the complex nature of phase
determination.

This study leverages a unique regional-scale radar disdrometer network coupled with
precipitation observations to develop a probabilistic phase partitioning model. The probabilistic
model follows a phase-guided partitioning (PGP) in the form of a chain of random forest models.
The precipitation phase is classified before partitioning to accurately replicate its intricate
behavior and to take advantage of the significant amount of validation data available through
such a network. Additionally, multiple PGP models with lower data requirements are developed
to evaluate the possibility of utilizing such models in practical operations. This study begins with
a description of the precipitation dataset and hydrometeorological variables used, followed by
the methodology used to develop the PGP models. Finally, the results section presents an
analysis of the dataset and evaluates the model's phase classification and partitioning
performance, comparing it to benchmark models of differing levels of complexity.

## 2  Data

### 2.1  Surface hydrometeorological measurements

The disdrometer network used for this study was deployed on the north shore of the St. Lawrence
River in the province of Quebec, Canada (Figure 1). It is part of a larger hydrometeorological
observation network operated by Hydro-Quebec, a public utility responsible for the generation
and distribution of electricity in Quebec. The network lies between latitudes 47.23 and 52.13 °N,
and longitudes 63.17 to 75.29° W, spanning an area of roughly 1,138,00 km². The 27 stations
have been in operation for varying periods of time between the years 2019 and 2023, totaling 80
site-years. The sites' elevation ranges from 315 to 641 m above sea level (ASL), for an average
of 469 m ASL. The names, coordinates, and operational timeframes for each station are
presented in the Appendix A.

The sites have a mean annual 2-m air temperature of 0.2°C and a mean annual cumulative
precipitation of 902 mm, calculated from 160 site-years of daily observations. Figure 2 illustrates
the distribution of annual mean daily 2-m air temperature and precipitation, as well as the
elevation of the sites. The annual precipitation decreases with latitude, as the northern sites (>51°
N) experience an annual mean of 813 mm. The southernmost sites (<48° N) see significantly
more precipitation, with an annual mean of 1002 mm. The sites follow a mostly normal
distribution in the 400 m range between sites, with elevations generally increasing northward.
Following Köppen climate classification, the sites are nearly evenly split between humid



continental (Dfb) and humid subarctic (Dfc) climates. The study period spans from October 1 to

June 1 of the following year, as the chances of snowfall are practically non-existent outside of these dates for the domain of interest.

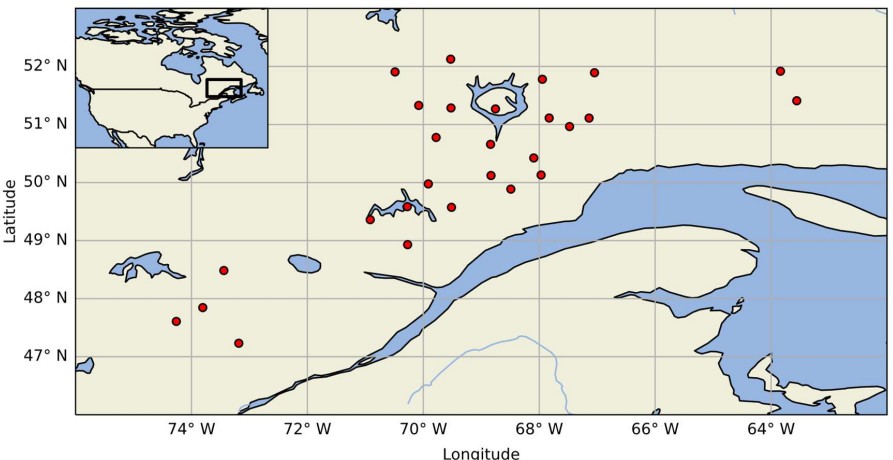

**Figure 1: Location of the study sites in eastern Canada. The black square in the top left inset corresponds to the map domain.**

Each site is equipped with a radar-based disdrometer (model WS100, Lufft), providing 15-min phase observations. The WS100 is a K-band (24 GHz) Doppler radar that classifies droplets into 11 size classes between 0.3 and 5.0 mm. The disdrometer assigns World Meteorological Organization (SYNOP $W_aW_a$ code) weather codes for no precipitation (code 0), rain (code 60), freezing rain (code 67), mix of snow and rain or drizzle (code 69), and snowfall (code 70). The

precipitation phase is identified according to the hydrometeor diameter-fall velocity relationships for water droplets outlined in Gunn and Kinzer (1949), as well as in Locatelli and Hobbs (1974) for solid-phase precipitation particles. Rain and snow falling velocity as a function of measured reflectivity from K-band Doppler radar was investigated in Atlas et al. (1973) and remains an area of active research (e.g. Garcia-Benadi et al., 2020; Kneifel et al., 2011; Löffler-

Mang et al., 1999; Sarkar et al., 2015). In addition, each site also provides measurements of SWE using a passive gamma ray monitoring system (model CS725, Campbell Scientific) and snow depth using an ultrasonic sensor (SD, model SR50A, Campbell Scientific).

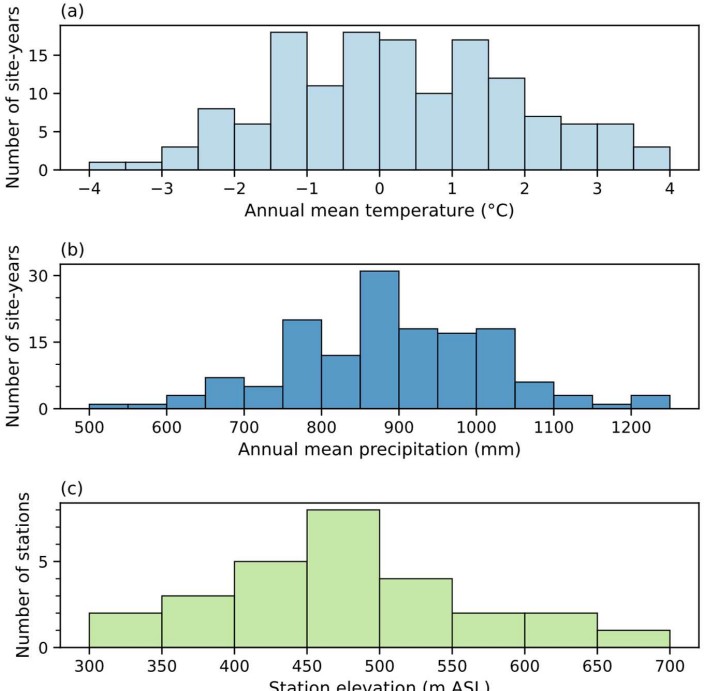

**Figure 2: Distributions of the (a) annual mean temperature, (b) annual mean precipitation and (c) elevation at**
**the study sites.**

The meteorological observations in this study are from weather stations near the disdrometer
stations, operated by Hydro-Quebec and SOPFEU, the province's wildfire prevention
organisation. The weather stations provide 15-min accumulated precipitation (model Pluvio² by
OTT, equipped with a single-Alter shield), allowing the coupling of precipitation with
concurrent disdrometer phase identification. The weather stations measure hourly air
temperature (model CS109, Campbell Scientific) and relative humidity (model HMP155a,
Campbell Scientific) sensors mounted at 2 m above ground level. Additionally, wind speed and
direction are monitored at a 15-min interval with a ground propeller anemometer (model 05103,
Young) mounted 2 m and 10 m above the ground surface.

Most of the study sites and weather stations are located in close proximity to each other.
Specifically, 67% of the study sites are within 3 km of or collocated with the nearest weather
station. The remaining stations are between a median distance of 7 km and a maximum distance
of 12 km. The only exception is the AUXLOUPS station and the nearest weather station, which
are separated by 28 km. To account for the elevation differences between the study sites and
weather stations, the air temperature measurements were adjusted using international standard
atmosphere methods. The discussion section will address the uncertainty related to the distance





between the study site and the weather station. The temporal resolution and detailed specifications about the study sites' instruments are provided in

Table 1.

**Table 1: Study site instruments details.**

| Model, manufacturer | Temporal resolution | Observation | Specifications |
|---|---|---|---|
| WS100, Lufft | 15 min | Precipitation phase | See section 3.1 and Appendix B. |
| Pluvio$^2$, OTT | 15 min | Precipitation rate | Resolution: 0.1 mm<br>Accuracy: ±0.05 mm |
| CS109, Campbell Scientific | 60 min | Near-surface air temperature | Accuracy: ±0.2°C (from 0°C to 70°C), increasing to ±0.5°C at −50°C |
| HMP155a, Campbell Scientific | 60 min | Near-surface relative humidity | Accuracy: ±(1.0 + 0.008 × reading) % RH (from −20°C to 40°C) |
| 05103, R.M. Young | 15 min | Wind speed 2 m above ground | Wind speed threshold: 1.0 m s$^{-1}$<br>Accuracy: ±0.3 m s$^{-1}$ |
| SR50A, Campbell Scientific | 60 min | Snowpack depth | Resolution: 0.25 mm<br>Accuracy: ±1 cm or 0.4% of distance to target |
| CS725, Campbell Scientific | 6 h | Snowpack SWE | Resolution: 1 mm<br>Accuracy: ±15 mm (from 0 mm to 300 mm) |

### 2.2    Reanalysis products

Hourly atmospheric data from the ECMWF-ERA5 reanalysis (Hersbach, 2023) are added to this study's dataset, to account for the energy transfer to falling hydrometeors in the atmosphere closest to the surface. Furthermore, this will help assessing the potential performance gain of incorporating gridded data, despite the spatial scale discrepancy with local observational data. The added data include temperature profiles for pressure levels of 1000 and 850 hPa. The

corresponding geopotential height of these levels is also added to the dataset. The values from the nearest 0.25° × 0.25° grid cell, roughly 28 km × 18 km at the study sites' latitude, are assigned



to every study site. Additionally, the hourly surface atmospheric pressure from ERA5-Land (Muñoz Sabater, 2019) is added to the dataset, as it was not measured at the weather stations used in this study. The atmospheric pressure from the nearest 9 km × 9 km grid space is assigned to every study site. From this data, the thickness $\Delta z$ between the 1000 and 850 hPa layers (m) is calculated with:

$$\Delta z = \frac{z_{850} - z_{1000}}{g} \qquad (1)$$

Where $z_{850}$ and $z_{1000}$ correspond to the geopotential heights (m$^2$ s$^{-2}$) at the top and bottom of the layer, and $g$ is the gravitational acceleration (9.81 m s$^{-2}$). The layer thickness between the two pressure levels is correlated with the mean temperature of the layer and indicates the travel time of the hydrometeor in the air column. It is also a commonly used variable in operational meteorological models (Feiccabrino et al., 2015). The pressure levels were selected based on their successful use in the classification of the precipitation phase at the surface in prior studies (e.g. Bourgouin, 2000; Shin et al., 2022). The temperature lapse rate $\Gamma$ (°C km$^{-1}$) is also calculated:

$$\Gamma = -\frac{\Delta T}{\Delta z} \times 1000 \qquad (2)$$

Where $\Delta T$ correspond to the temperature difference between the 850 and 100 hPa layers (°C).

## 3 Methodology

### 3.1 Precipitation data processing

The observed 15-min precipitation amounts were compiled at the hourly timestep. Each 15 min precipitation data segment was coupled with a disdrometer phase identification. Both valid, non-zero values were required for the data segment to be included in the analysis. A first filter was applied, where hourly precipitations rates < 0.2 mm h$^{-1}$ were considered erroneous trace amounts, following Environment and Climate Change Canada methodology (ECCC, 2024; Chartrand et al., 2023; Marinier et al., 2023). A second filter was also applied where precipitations rates > 110 mm h$^{-1}$ were considered erroneous (Smith et al., 2022). A neutral aggregating filter (Ross et al., 2020) was then applied to eliminate noise and diurnal oscillations to the precipitation data. Additionally, hourly precipitation exceeding 30 mm h$^{-1}$ was visually inspected. Any data not consistent with nearby stations was considered invalid.

The disdrometers used in this study can identify freezing rain and a mix of rain and snow in addition to snow and rain. However, as most hydrological models only interpret the effect of snow and rain, this study focuses on the prediction of solid and liquid precipitation. Therefore, the disdrometer identifications of freezing rain and of mix of snow and rain/drizzle were





aggregated with snow and rain events, respectively. The selected phase aggregation aims to
group the phases that are most similar in terms of hydrological influence and average occurring
temperature. These assumptions are supported by an analysis of the effect of each precipitation
phase on the snowpack properties (height and snow water equivalent), as detailed in Appendix
B.

When solid precipitation was identified, the universal transfer functions of Kochendorfer et al.
(2017) were applied to adjust for wind-induced gauge undercatch. To do so, local hourly wind
speed and temperature measurements at gauge height were used, which were shown to provide
appropriate corrections for sites in boreal climates (Pierre et al., 2019). The solid and liquid 15-
min precipitation were then compiled at hourly time steps and partitioned into liquid and solid
precipitation fractions, totaling 44,790 data points. The resulting phase partitioning was used to
classify the phase of each precipitation event as solid, liquid, or mixed, with respective dataset
proportions of 71%, 22% and 7%.

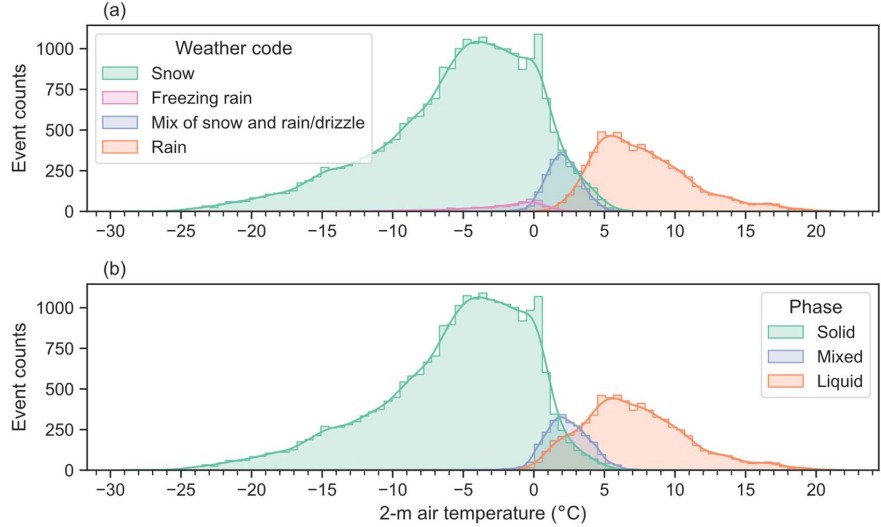

**Figure 3: Hourly event counts of (a) the main precipitation phase identified by the disdrometers and (b) the
aggregated precipitation phases according to the 2-m air temperature.**

Figure 3 shows that the aggregation mainly affects mixed and liquid precipitation, and that the
aggregation of very few freezing rain events with snow events results in solid precipitation
counts being similar to snowfall counts. The aggregation of the mix of snow and rain/drizzle
with rain results in an increase in liquid precipitation in the 0-5°C range. Mixed-phase
precipitation occurs in the same air temperature range as that of mix of snow and rain/drizzle,
suggesting that this phase is often present in mixed-phase precipitation, thus validating the
aggregation. A cursory analysis of the mixed-phase precipitation events revealed that events



with a phase transition between snow and mix of snow and rain/drizzle account for roughly 75% of the mixed-phase events. Transitions from rain to snow are infrequent and represent roughly 15% of the mixed-phase precipitation events, while the remainder includes other phase

combinations.

## 3.2   Model performance evaluation

The models presented in this study are evaluated for their ability to correctly predict the precipitation phase using a variety of performance metrics. First, the metrics used to quantify the predictive ability of the models are the precision (PRE) and the recall (REC), as well as the

F1 score. The combination of precision and recall is commonly used to evaluate model classification performance, as the metrics indicate different information. The precision indicates the proportion of correct predictions for a given phase, while the recall indicates the hit rate for a given phase. Consequently, model precision and recall are inversely proportional. Therefore, a model that achieves good performance in both metrics is desirable.

$$PRE = \frac{TP}{TP + FP} \qquad (3)$$

$$REC = \frac{TP}{TP + FN} \qquad (4)$$

Where $TP$, $FN$, $FP$ are the true positive, false negative and false positive counts respectively for

a given phase. The F1 score, being the harmonic mean of the precision and recall, is a useful metric to quantify the general performance of the model, as it harshly penalizes a poor score in either metric.

$$F1 = 2\left(\frac{PRE \times REC}{PRE + REC}\right) \qquad (5)$$


These metrics are computed for each precipitation phase separately. A general score is also computed by weighing each phase's score according to its proportion in the dataset. As such, the weighted F1 score is used as a general classification performance metric, as it combines both precision and recall, and harshly penalizes a poor score in either of them while also considering

the dataset imbalance.

Second, the model partitioning performance model are evaluated based on the predicted solid and liquid precipitation amounts. The metrics used are the coefficient of determination $R^2$ and the RMSE. Due to the slightly asymmetric phase distribution and overlap between the phases

shown in Figure 3, different $R^2$ are calculated for the solid and liquid precipitation. Thereby, the



metrics are calculated on the phase-separated precipitation rather than on the precipitation fraction, as the precipitation phase could be solid, liquid or mixed for a given temperature. However, the RMSE is only calculated for the solid precipitation, as the score would be equal for both solid and liquid precipitation. In other words, the RMSE amounts to the root mean squared misclassified precipitation.


Finally, the partitioning performance metrics are performed on different subsets of the dataset using a K-fold method. The K-fold validation method is commonly used to assess the variability of model performance with machine learning methods. By using different subsets of the dataset to train and validate the model K times, a more general performance can be assessed. Because of the fewer liquid and mixed precipitation events compared to solid precipitation events, the K-Fold is also stratified to maintain phase proportions between training and validation sets from fold to fold. As such, in the case of the partitioning validation, the variability of the precipitation amounts from fold to fold must be considered. The performance metrics are repeated until the variance of the partitioning performance metrics stabilizes. In this case, the validation was performed with 5-fold validation and was repeated six times for a total of 30 validation folds.



### 3.3 Phase-guided probabilistic precipitation phase model

Machine learning algorithms are powerful tools for building classification and regression models. Random forests (Breiman, 2001) are commonly utilized in the environmental sciences due to their simple implementation and lower susceptibility to overfitting compared to other models. The model is based on decision trees, where variables are randomly chosen at each node to create a prediction. Therefore, the decision trees, each unique due to randomness, provide predictions that are ultimately aggregated to generate a final well-informed prediction.


Given the overlapping phases of the data set, a Random Forest (RF) classifier is used to predict the precipitation phase with a probabilistic approach. The procedure to develop the RF model is illustrated in Figure 4. To address the phase type imbalance, the data were adjusted by undersampling the solid phase and increasing the weight of both liquid and mixed precipitation phases in the dataset. To achieve this, only data points with air temperatures between −4 and 8°C were kept in the analysis. The phase proportions resulting from the undersampling are 60% solid, 26% liquid and 14% mixed. The data were then split using an 80/20 ratio between the training and validation sets respectively, resulting in 13,339 data points for training and 3,335 for validation. Because solid precipitation events make up most of the samples, the training and validation sets were stratified to maintain the same phase distribution between the two subsets. Hyperparameters were optimized on the training set to increase the model performance and reduce the chance of overfitting by using a stratified 5-fold cross validation and maximizing for







a weighted F1 score. The RF classifier was then retrained on the entire training data set and was
       ready for use on the validation set.

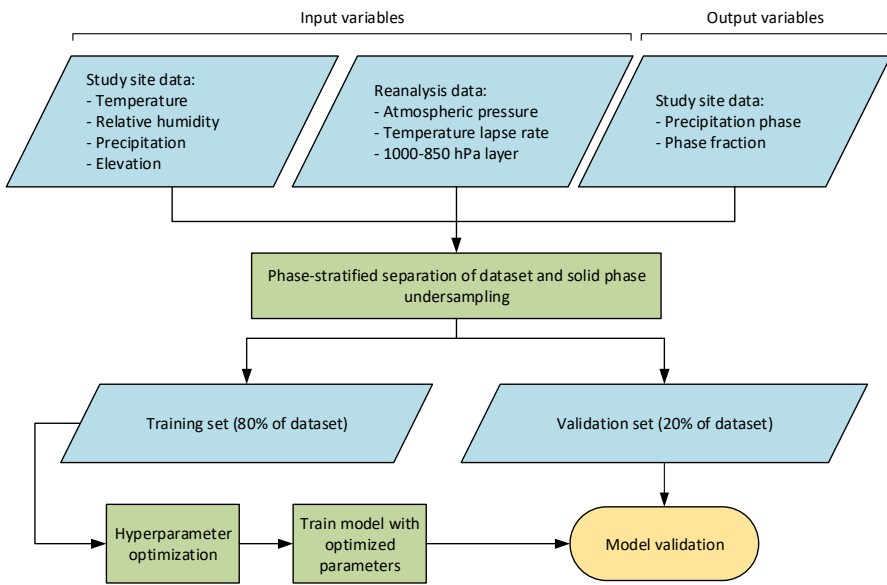

**Figure 4: Development and validation methodology for the Phase-Guided-Partitioning model.**

       While the precipitation partitioning is straightforward when the predicted phase is either solid
or liquid phase, it is less so for predicted mixed phase, where a solid and liquid precipitation
       fraction must be assigned. Thus, in the case of a predicted mixed phase, an RF regression model
       is developed following the same steps described above. The loss function used to optimize the
       regression model parameters is the mean-squared error (MSE), to increase the penalty on larger
       errors. The Phase-Guided-Partitioning model predicts a precipitation phase, as well as a solid
and liquid precipitation partitioning according to the predicted phase, with the complete process
       illustrated in Figure 5.



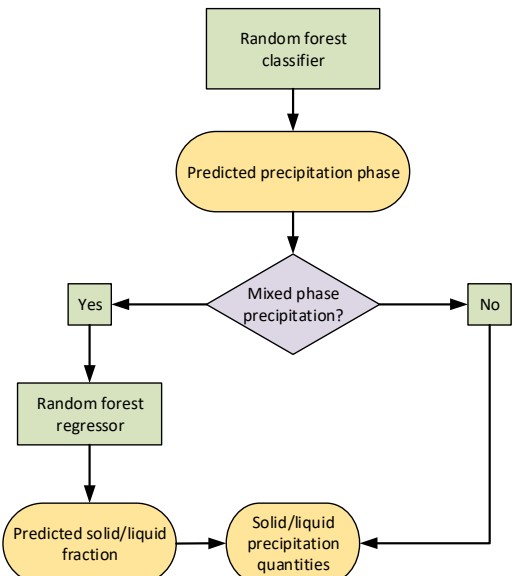

**Figure 5: Phase-Guided-Partitioning model structure.**

Multiple PGP models using a combination of atmospheric variables were developed. The subsets

of input variables of the PGP models accommodate different levels of data availability, ranging
from the strictest minimum data requirements (e.g., in an operational context) to atmospheric
variables, with each subset fully incorporating the previous subsets (see Table 2). This approach
will help to quantify the impact of some atmospheric variables that are not measured at surface
weather stations. The simplest model, PGP_basic, includes only 2-m air temperature and site

elevation. Next, PGP_hydromet includes all related near-surface hydrometeorological data, such
as relative humidity, atmospheric pressure, and precipitation rate. Finally, the PGP_full model,
as discussed in the previous sections, incorporates atmospheric data from reanalysis, specifically
the thickness of the 1000-850 hPa layer and temperature lapse rate.

**Table 2: Listing of the input variables used in the tested PGP models.**

| PGP model | Input variables |
|---|---|
| PGP_basic | 2-m air temperature, elevation |
| PGP_hydromet | 2-m air temperature, elevation, relative humidity, atmospheric pressure, precipitation rate |
| PGP_full | 2-m air temperature, elevation, relative humidity, atmospheric pressure, precipitation rate, 1000-850 hPa layer thickness, temperature lapse rate |




### 3.4 Benchmark phase partitioning models

The benchmark models used for this study are common methods of increasing complexity found in hydrological models. First, a single 2-m air temperature threshold (ST) model is used as a baseline comparison. This model separates precipitation into solid and liquid phases based on a calibrated air-temperature threshold. While this type of model is widely used, it is generally associated with larger partitioning errors on a seasonal basis (Harpold et al., 2017). Second, a linear transition (LT) model is used. It allows for mixed-phase precipitation, while still being of low complexity. LT partitions the solid and liquid precipitation according to a linear relationship between a snow and rain temperature threshold. Finally, the psychrometric energy balance (PB) model is used, which is a physically based phase partitioning method that integrates the relative humidity to estimate the hydrometeor temperature (Harder and Pomeroy, 2013). The estimated hydrometeor temperature is then used as an input in a two-parameter curvilinear relationship. All three benchmark models are calibrated individually with the least squares method, minimizing the error with the observed solid-phase fractions. The models and their calibrated values are given in Appendix C. The precipitation phase can then be inferred from the predicted fractions. Probabilistic models from previous studies (e.g. Behrangi et al., 2018; Jennings et al., 2018), were not included as benchmark models as the omission of mixed-phase events in such studies makes it difficult to compare the results.

### 3.5 Input variable importance analysis

A common way to interpret input variable importance for a machine learning model is to use permutation importance, which helps decreasing the black-box aspect of machine learning algorithms (Mcgovern et al., 2019). The performance of the model is computed according to a chosen scoring scheme. Each variable of the model is then shuffled individually. The goal of this step is to break the relationship between a variable and the desired prediction. After each shuffle, a performance score is calculated to show the decrease in model performance. This process is then repeated several times to account for data variability. Thus, the relative importance of each input variable to the model can be quantified with the resulting performance decrease. Permutation importance analysis provides only the importance of an input variable to the model, not the inherent information provided by that variable. However, when shuffling a variable that is highly correlated to another, the model can still find the shuffled variable's information when performing permutation importance analysis. In practice, this is an important consideration as it means the importance of either, or both, input variables can be lower. This analysis offers insight into the crucial variables for the PGP models and how they can be further improved.



## 4   Results

### 4.1   Dataset analysis

The distribution of the hydrometeorological variables categorized by precipitation phase are displayed in Figure 6. The temperature distributions show a significant overlap between all three phases from 1.5 to 3.6°C, similar to that reported in Jennings et al. (2023). Mixed precipitation probability peaks at approximately 2.4°C. The distribution of relative humidity reveals that precipitation is associated with near liquid water vapor saturation conditions, with a median value of 97%, regardless of the precipitation phase. The mean precipitation rate is generally low, at 0.9 mm h$^{-1}$. The median precipitation rate for mixed-phase events is generally the highest at 0.8 mm h$^{-1}$, followed by liquid-phase events at 0.7 mm h$^{-1}$ and solid-phase events at 0.6 mm h$^{-1}$. Atmospheric pressure distributions are similar for both liquid and mixed-phase precipitation events. The mean air pressure during the solid precipitation events is comparable to that of the other phases, but there are more events between 90 and 92 kPa. The distribution for the thickness of the 1000-850 hPa layer closely mirrors that of air temperature, given their general correlation. The temperature lapse rate averages 4.9°C km$^{-1}$ and distributions, especially solid precipitation, show a bias toward the standard atmospheric lapse rate of 6.5°C km$^{-1}$.

The overlap of the phase distributions for each input variable, most notably the air temperature, indicates that a probabilistic approach is appropriate for predicting the precipitation phase. Indeed, between approximately 0 and 4°C, solid and liquid precipitation may occur separately or coexist. According to findings in previous studies, precipitation over land is more likely to occur in a single phase than in mixed phase precipitation (Dai, 2008; Froidurot et al., 2014), as is the case in this study, where only 13% of the precipitation data points are mixed phase. There is, however, a narrow 2-m air temperature range, between 2 and 2.5°C, where mixed-phase probability exceeds the probability of single-phase precipitation. An appropriate phase partitioning model must thus accurately predict the phase in the temperature interval where solid, liquid, and mixed precipitation occurrences overlap while also providing accurate partitioning when needed.

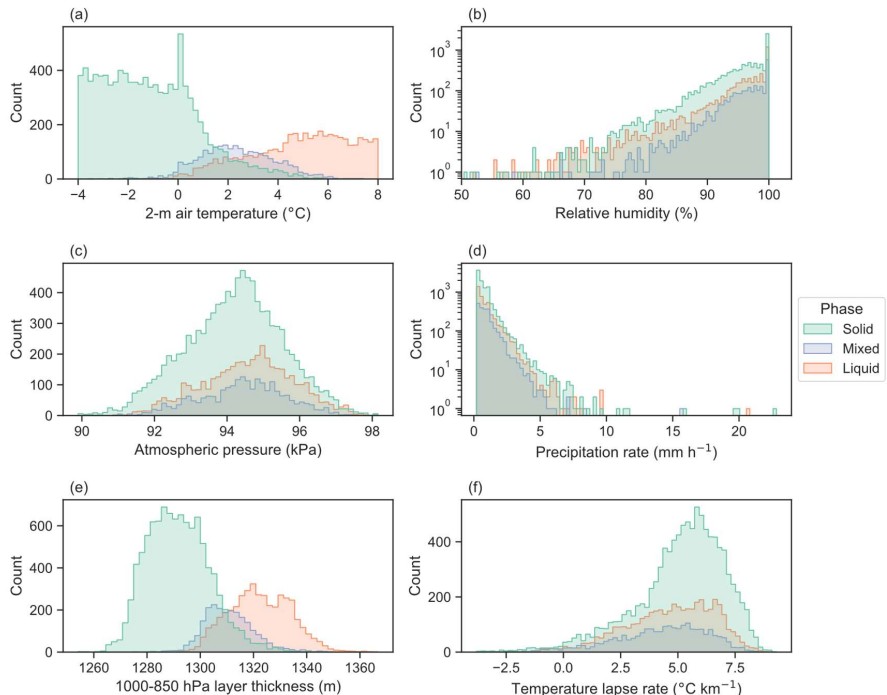

**Figure 6: Input distributions separated by phase of (a) 2-m air temperature, (b) relative humidity, (c) atmospheric pressure, (d) precipitation rate, (e) 1000-850 hPa layer thickness, (f) air temperature lapse rate.**

## 4.2 Phase classification

Figure 7 shows the phase density distribution of the benchmark models and the PGP models. The phase density distributions show the limitations of benchmark phase partitioning models, namely that the mixed phase is absent or overrepresented compared to the observations. Due to the relationships used to create the benchmark models, the overlap between all three phases is not accurately represented. By including relative humidity, PB can model phase overlap, but this

does not improve the modelled phase distributions density with respect to the observations. The PGP models reproduce the observed phase overlap well, but slightly overpredict the mixed phase, affecting both the solid and liquid-phase predictions. PGP_basic overpredicts the most the mixed phase, while the difference between PGP_hydromet and PGP_full is marginal. This result suggests possible improvements to PGP models, particularly for mixed-phase

precipitation.





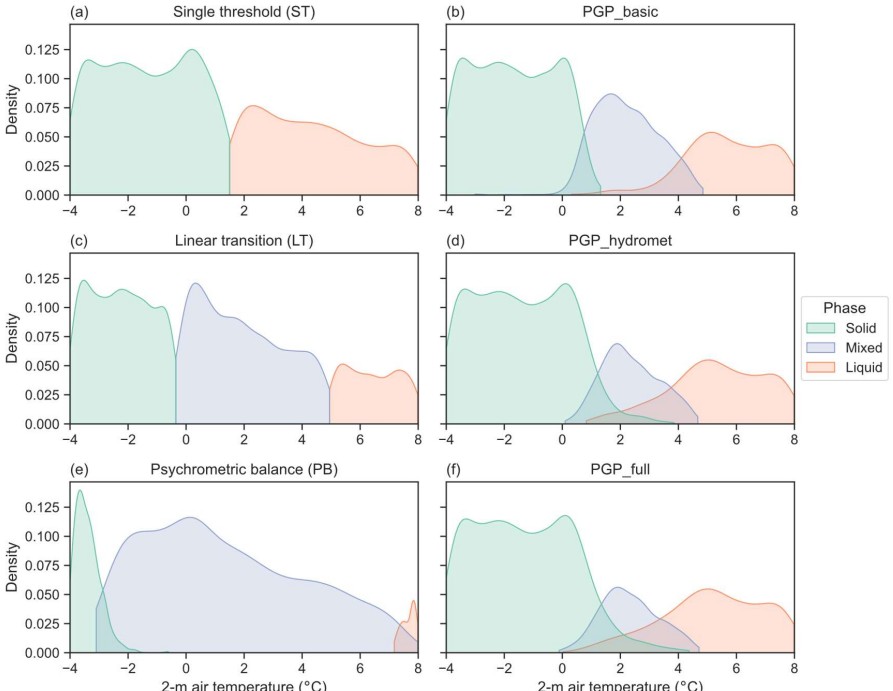

**Figure 7: Modelled phase distributions according to 2-m temperature of the (a) single threshold, (b) PGP_basic, (c) linear transition, (d) PGP_hydromet, (e) psychometric balance and (f) PGP_full. PGP model details are summarized in Table 2.**

The classification scores of the various models in Table 3 were weighted to reflect the precipitation phase proportions in the dataset and reveal insightful performance patterns. ST performs well in all three metrics due to the low likelihood of mixed phase occurrence. When evaluating the overall classification performance using the F1 score, LT follows ST because of

465 a disparity between precision and recall that affects its F1 score. The lower recall score for LT can be attributed to its overprediction of the less frequent mixed phase, which, in turn, negatively affects the recall of other phases. This enhances the model's weighted precision by decreasing the number of false positives in non-mixed-phase prediction. The same reasoning can be more extensively applied to PB's weighted scores. The mixed phase's overlap with other phases

significantly decreases the model's overall recall. Finally, the weighted F1 score for the PGP models shows that they have a more robust general performance, as they have high weighted precision and recall scores, while having a small disparity between both scores.





**Table 3: Weighted classification scores for Single Threshold (ST), Linear Transition (LT), Psychrometric Balance (PB) and Phase-Guided Partitioning PGP models. PGP model details are summarized in Table 2.**

| Model | F1 | Precision | Recall |
|---|---|---|---|
| ST | 0.74 | 0.71 | 0.79 |
| LT | 0.71 | 0.88 | 0.66 |
| PB | 0.31 | 0.88 | 0.30 |
| PGP_basic | 0.82 | 0.86 | 0.80 |
| PGP_hydromet | 0.84 | 0.85 | 0.83 |
| PGP_full | 0.84 | 0.84 | 0.84 |

The phase-separated classification metrics provide further insight into the performance of the models, shown in Figure 8. The F1 score provides an overall performance for each phase prediction. PGP_full has the best F1 scores for both the solid and liquid phases, while PGP_hydromet has a higher F1 for the mixed phase. PGP_basic is generally the third best performing model in terms of F1 score, except for the solid phase, where ST outperforms it. While it is not able to predict the mixed phase, ST has the highest scores for the liquid and solid precipitation phases out of the benchmark models. This is probably because mixed-phase precipitation events are only roughly 13% of the samples, this low proportion does not significantly decrease the model's performance. LT performs slightly worse than ST for both solid and liquid phases F1 scores but has the highest mixed-phase F1 score out of the benchmark models.

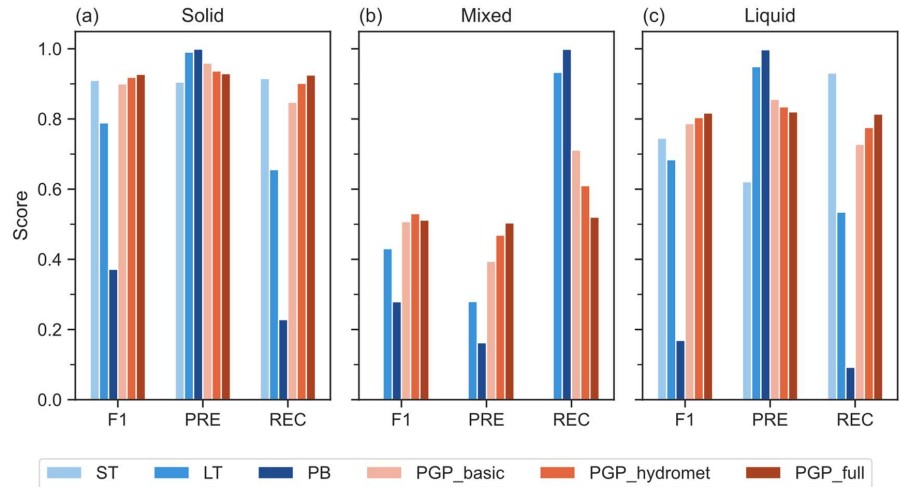

**Figure 8: Model phase classification metrics separated by (a) the solid, (b) the mixed and (c) the liquid phase. PGP models. PGP model details are summarized in Table 2.**





PB's poor F1 scores are explained by the overlaps between the phases shown in Figure 7. The model allows the predicted mixed phase to overlap with the predicted solid and liquid phases, which is the opposite behavior of the observed phase density, where mixed-phase precipitation mostly exist in the solid and liquid phase overlap. Given the modelled phase density and the resulting classification scores, the phase prediction ability of both LT and PB suffers from overprediction of the mixed phase. This is evidenced by the significant disparity between the precision and recall scores of LT and PB for the mixed phase. A high recall score signifies that the model minimizes the number of false negatives, which negatively affects the model's precision. Thus, overpredicting the mixed phase greatly reduces the models' precision for the mixed phase, while greatly increasing their recall for the mixed phase. Conversely, the conservative prediction of the liquid and solid phases increases the precision of the model but decreases the recall for both phases.

Although they are not always the best models in terms of either precision or recall, the PGP models have the best general performance, making them more reliable for phase prediction. Thus, PGP models significantly reduces phase identification error by showing high precision and recall, with small disparities for solid and liquid phase prediction. However, PGP's main distinguishing feature is its general ability to predict mixed-phase precipitation. Furthermore, the disparity between the precision and recall scores for the mixed phase is much smaller than for the other models studied, indicating that the overprediction of the mixed phase is much less severe for PGP.

### 4.3 Precipitation partitioning

The average regression metrics in

Figure 9 displays how the regression metrics vary across validation folds. The precipitation rate variability has a significant impact on ST's performance, making its ability to partition precipitation highly variable from winter to winter. In contrast, LT and PB exhibit better performance than ST due to their ability to partition the solid and liquid phases, with much less variability in performance. The variability of R2 for liquid precipitation is lower for LT and PB than for solid precipitation, because fewer of these events occur. For all regression metrics, LT and PB have similar performance. This is most likely due to the very humid environment, which decreases the difference between the 2-m air temperature and the hydrometeor temperature computed for PB.

Table 4 show the partitioning performance of the various models. All models have a high $R^2$ for solid precipitation, likely due to the abundance of solid precipitation. However, model performance decreases for liquid precipitation $R^2$, with ST being significantly lower than other models. This trend is also observed for the RMSE. While ST is the worst performing model, LT,





PB and PGP_basic perform similarly in all regression metrics. The inclusion of hydrometeorological data in PGP_hydromet leads to a slight increase in performance. Lastly, the inclusion of atmospheric data in PGP_full improves performance compared to the other models.

Figure 9 displays how the regression metrics vary across validation folds. The precipitation rate variability has a significant impact on ST's performance, making its ability to partition precipitation highly variable from winter to winter. In contrast, LT and PB exhibit better performance than ST due to their ability to partition the solid and liquid phases, with much less variability in performance. The variability of $R^2$ for liquid precipitation is lower for LT and PB

than for solid precipitation, because fewer of these events occur. For all regression metrics, LT and PB have similar performance. This is most likely due to the very humid environment, which decreases the difference between the 2-m air temperature and the hydrometeor temperature computed for PB.

**Table 4: Average regression scores for Single Threshold (ST), Linear Transition (LT), Psychrometric Balance**
**(PB) and Phase-Guided Partitioning (PGP) models. PGP model details are summarized in Table 2.**

| Model | $R^2$ solid | $R^2$ liquid | RMSE (mm) |
|---|---|---|---|
| ST | 0.76 | 0.65 | 0.40 |
| LT | 0.86 | 0.80 | 0.31 |
| PB | 0.86 | 0.80 | 0.31 |
| PGP_basic | 0.87 | 0.81 | 0.30 |
| PGP_hydromet | 0.88 | 0.83 | 0.29 |
| PGP_full | 0.89 | 0.85 | 0.27 |

Generally, the performance of PGP_basic is similar to that of LT and PB, with slight differences. PGP_basic is more variable in its performance for solid precipitation $R^2$ and RMSE. This variability can be attributed to the misclassification of precipitation events due to its limited

input variables. The $R^2$ scores for PGP_hydromet are less variable than for PGP_basic, while its RMSE is the most variable out of the PGP models. PGP_full exhibits the lowest variability for both $R^2$ and is the only PGP model with RMSE variability similar to benchmark models LT and PB.

The broader RMSE score range of the PGP models highlights the impact of misidentified phases.
Misidentification can be more costly than for a benchmark model that systematically separates precipitation into solid and liquid phases for temperatures where mixed-phase events are possible. Furthermore, models such as LT and PB achieve partial accuracy in phase partitioning by forcing mixed-phase precipitation, but if a PGP model misclassifies the phase, the entire





precipitation event may be incorrectly partitioned. However, PGP models do show that phase identification prior to phase partitioning can reduce the overall error of a model for both solid and liquid precipitation. This suggests that improved phase identification, specifically with mixed-phase prediction, could greatly enhance the accuracy of precipitation partitioning from PGP model.

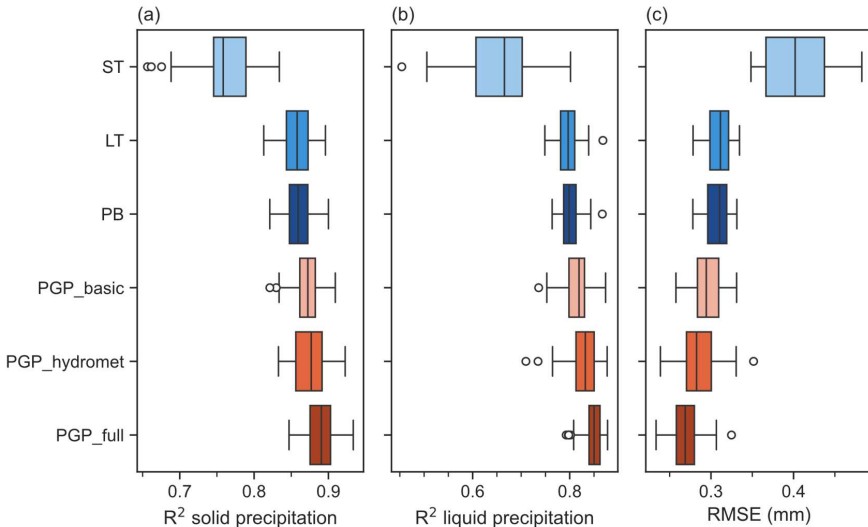


**Figure 9: Model regression performance in (a) $R^2$ for solid precipitation, (b) $R^2$ for liquid precipitation and (c) RMSE. PGP model details are summarized in Table 2.**

### 4.4 Input variable importance

Figure 10 shows the correlation matrix of PGP_full input variables. While the correlation of
most input variable combinations is low, the 2-m air temperature and 1000-850 hPa layer thickness are highly correlated. The layer thickness is affected by environmental temperatures, as the air density is inversely proportional to its temperature, which increases the distance between two pressure levels. There is a moderate negative correlation between elevation and air pressure, probably because of the small range of study site elevations. The temperature lapse
rate has a small correlation with almost all features.

Improving PGP models' ability to accurately predict the mixed phase is manifold. First, the PGP models tend to overpredict the mixed phase, which also negatively impact their ability to predict the other phases. In turn, this also affects the models' partitioning error. For these reasons, the chosen scoring scheme for the permutation importance is the weighted F1 score, to consider the
imbalanced proportions of the phase data.



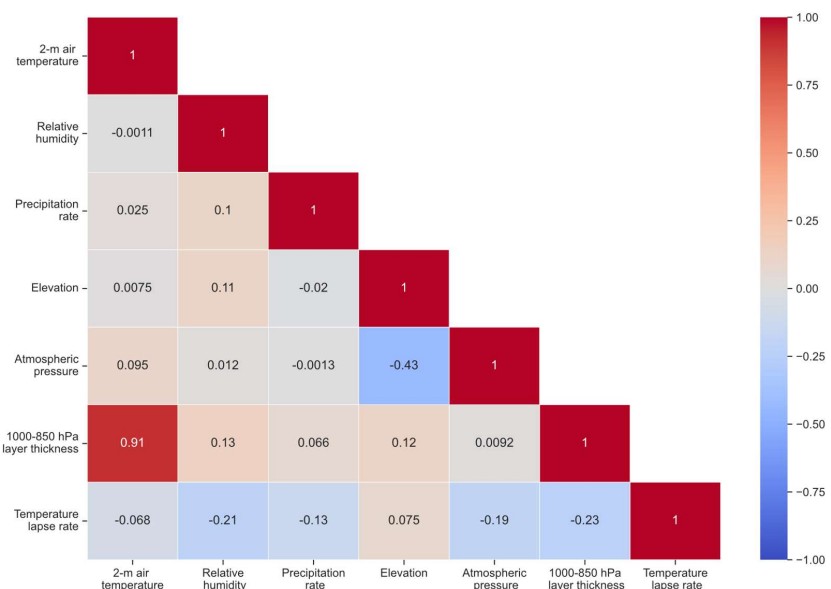

**Figure 10: Correlation matrix of PGP_full input variables pairs.**

Figure 11 shows the permutation importance of PGP_full input variables and the resulting

decrease in the weighted F1 score on the validation set. The 2-m air temperature is the most

important variable, with its permutation decreasing the score by more than 0.2. The second most

important variable is the 1000-850 hPa layer thickness, a result shared by Shin et al. (2022).

Because of its high correlation with the 2-m air temperature, it is difficult to interpret the real

importance of this variable. In terms of classification performance, the addition of this variable

seems to provide small improvements, as shown by the differences in classification metrics

between PGP_hydromet and PGP_full.

For the remaining variables, the importance decreases sharply. However, while the individual

importance of the variables is low, they improve the phase classification when combined. This

suggests that the additional variables used in PGP_full likely improve mostly mixed-phase

prediction, which is supported by the model's performance in section 4.2. The elevation is used

to approximate the atmospheric pressure of a site and can improve phase partitioning (e.g. Ding

et al., 2014; Behrangi et al., 2018). Furthermore, atmospheric pressure is often cited as an

important variable for phase partitioning (e.g. Behrangi et al., 2018; Dai, 2008; Jennings et al.,

2018). The thinner air in low-pressure environment allows snow to reach the ground faster. The

temperature lapse rate provides key information regarding the amount of energy the

hydrometeors absorb before reaching the ground.

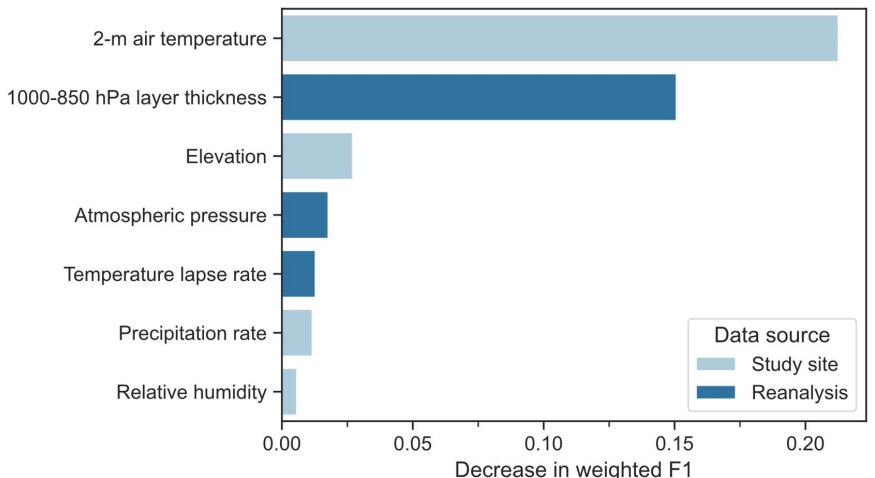

**Figure 11: Permutation importance of PGP_full input variable, showing the decrease in the model's weighted F1 score.**

The precipitation rate has a minor impact on the performance of the model. Nevertheless, it may hold significance for the prediction of the mixed phase. The precipitation rate is linked to the precipitation phase as it increases the energy required to completely melt falling precipitation (Froidurot et al., 2014; Thériault et al., 2010). However, its effect is minimal, most likely due to the small proportion of mixed-phase events. Finally, the model ranks relative humidity as the

least important feature. This outcome is unexpected because relative humidity was shown to have a significant effect on phase partitioning (e.g. Behrangi et al., 2018; Jennings et al., 2018). One explanation could be the high percentage of data points near water vapor saturation, resulting in the variable being less regionally significant than more heterogeneous regions such as mountain ranges. Besides, this could account for the PB model's underwhelming

classification accuracy since it utilizes relative humidity to determine the precipitation phase and the fact that it was developed for the drier climate near the Canadian Rockies.

## 5    Discussion

### 5.1    Model performance and input variable importance

The classification and regression metrics of the PGP models show that phase classification prior
to phase partitioning reduces the partitioning error of solid and liquid precipitation, while also providing a more reliable phase prediction than benchmark models. The use of radar-based disdrometer measurements enabled the partitioning step of the model by providing precipitation fractions for mixed-phase events, a flaw mentioned in other studies (Froidurot et al., 2014; Jennings et al., 2018). Out of the benchmark models, ST displayed the best classification


performance, despite not allowing for mixed-phase precipitation. The tendency of overpredicting mixed-phase precipitation of both LT and PB reduced their overall classification performance. This general behavior was also observed in Leroux et al. (2023), where simpler methods outperformed methods based on a precipitation phase fraction. However, ST showed significantly worse partitioning performance compared to LT and PB. The limitations of

precipitation fraction-based models are highlighted by the fact that LT and PB were the worst performing phase classification benchmark models, despite being the best partitioning benchmark models. These models were calibrated to minimize partitioning error, but in doing so, they are biased toward predicting mixed-phase precipitation. As such, there is a trade-off between classification and partitioning error for precipitation fraction-based models such as LT

and PB.

PGP_basic, while showing an improvement in phase classification, did not significantly outperform the partitioning of benchmark models LT and PB. PGP_hydromet showed improved phase classification, notably for mixed phase, and partitioning. PGP_full showed further increase in overall performance, while also reducing the partitioning error variability. However,

all PGP models tended to over predict the mixed phase. Reducing the overprediction of mixed phase is a persistent challenge in improving precipitation phase modeling, as noted in previous studies (Casellas et al., 2021; Leroux et al., 2023).

The permutation importance analysis showed that most input variables used, apart from the 2-m air temperature, are of low importance. However, the classification performance improvement

of PGP_hydromet and PGP_full show the cumulative importance of the additional variables used, most notably for mixed-phase prediction. Despite many studies demonstrating its impact on precipitation phase, relative humidity was found to be the least important factor. This is likely due to the regional homogeneity, with most observations occurring near liquid water vapor saturation. The site elevation was considered important for phase classification, even though it

is a constant variable. This suggests that an atmospheric pressure estimated by the elevation could provide enough information relevant to improve phase classification. Still, out of the hydrometeorological variables, the atmospheric pressure had the most impact on phase classification performance. This is in line with other studies that found it has a significant impact on precipitation phase (e.g. Behrangi et al., 2018; Dai, 2008; Jennings et al., 2018), although

generally to a lesser extent than relative humidity, when considering the regional variability.

The precipitation rate's low importance is most likely because it affects mixed-phase prediction, thus has low impact for overall performance. According to Thériault et al. (2010), higher precipitation rates raise the likelihood of larger hydrometeors, which require more energy to melt. Consequently, there is an increased likelihood of mixed-phase precipitation occurring in

the form of partially melted hydrometeors. As noted by Feiccabrino et al. (2015), higher





precipitation rates can lead to snowfall happening at warmer temperatures due to the presence of unstable air below the isothermal layer.

The permutation importance 1000-850 hPa layer thickness is second to the 2-m air temperature. However, because of the high correlation of the pair of variables combined with the moderate classification improvement of PGP_full, the real importance of the 1000-850 hPa layer is most likely low. While the importance of the temperature lapse rate is low, the partitioning results demonstrated that incorporating gridded atmospheric variables alongside local observations led to a reduction in the variance of the regression performance. This finding is noteworthy because few studies, as pointed out by Harpold et al. (2017), have explored the impact of incorporating atmospheric reanalysis data into phase modeling. Froidurot et al. (2014) indicated that models using atmospheric data did not greatly improve the phase prediction, as is the case in this study's classification performance. Furthermore, Dai (2008) emphasizes the terrain-dependent nature of lapse rates. Thus, even though the study sites in the region are relatively similar in terms of terrain, the importance of lapse rates in the modeling process was still significant, contrary to the fairly homogenous relative humidity measurements.

### 5.2 Coupled precipitation data uncertainty

There are uncertainties regarding the results due to the dataset and assumptions employed. Hydrological models commonly limit the precipitation phases to solid or liquid. Nonetheless, this dataset includes a considerable quantity of mixed snow and rain/drizzle events, and it is uncertain how hydrological models should handle this precipitation phase. The phase aggregation step considered the behavior of the snowpack following the different phases detected by the disdrometers. However, phase identification errors have the potential to introduce uncertainties in the results. To measure this uncertainty, it is recommended that studies be conducted using collocated WS-100 disdrometers and other well-documented options such laser disdrometers, to assess the differences between ground-truth providing instruments (Harpold et al., 2017).

Another source of uncertainty arises from the coupling of precipitation amounts and phase observations. Fehlmann et al. (2020) demonstrated that laser disdrometers have low missed event and false alarm rates for sub-daily integration times compared to precipitometers, but no such study was conducted with the radar-based disdrometers of this study. Additionally, the study from Fehlmann et al. (2020) was carried out in a sheltered site from the wind, implying that the wind-induced gauge undercatch could not be studied. In turn, the wind could influence the missed event and false alarm rates of this study's instruments. In this study, data segments where either the precipitation gauge or disdrometer did not detect any precipitation were discarded. Figure 12 displays the hit-rate of both the instruments at the initial 15-min intervals and shows that the instruments' hit-rates are generally in agreement. The instrument hits were





normalized over the precipitation gauge observations to compute relevant agreement metrics. The precipitation gauge is considered as ground-truth as it would be used in conjunction with a precipitation phase model and phase observations are rare in operational context.

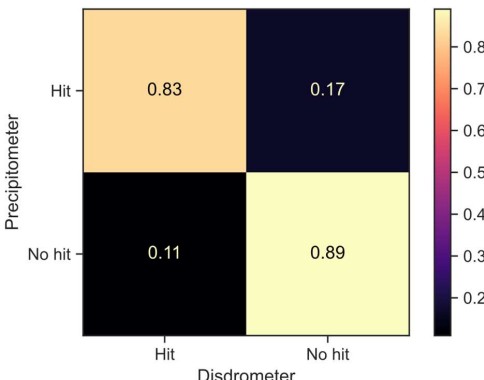


**Figure 12: Confusion matrix of the precipitometers and disdrometers 15-min precipitation hit-rate, normalized over the precipitometer observations. The upper-left metric is the probability of detection, upper-right metric is the miss rate, and the lower-left metric is the false alarm rate.**

Precipitation data segments of 0.1 mm coincide with 38% of the disdrometer misses. Assuming
that a significant portion of this precipitation data would be labeled as trace amounts when resampled at the hourly interval (< 0.2 mm), the probability of missed events is likely lower in reality. Multiple factors could account for disagreements between the instruments, including the effect of wind, which likely varies from instrument to instrument, and the fact that certain stations are not collocated or nearby. However, the environmental effects of disdrometer
performance lacks previous studies and requires more detailed investigation as outlined in Harpold et al. (2017).

Figure 13 shows the variation of the instrument agreement according to the distance between stations. The station pairs are divided into four distance categories: less than 3, 4 to 7, 7 to 12 and more than 12 km. Generally, the stations separated by less than 3 km show better agreement,
with a few outliers. However, the instrument agreement does not seem to decrease with distance, as the 4 to 7 km category exhibits the poorest agreement. Notably, the AUXLOUPS station, separated by 28 km with the nearest weather station, has a probability of detection of 0.79, a false alarm rate of 0.14 and a miss rate of 0.21. These instrument agreement metrics are only slightly worse than the metrics of the 15-min dataset in Figure 12. This suggests that instrument
agreement is linked to site specific conditions rather than distance between stations. However, by discarding data points where the instruments do not agree, we ensure that precipitation events are consistent across study sites and weather stations. In addition, the coupling of instruments



from nearby station brings the spatial scale of the observational data closer to the scale of the reanalysis data.

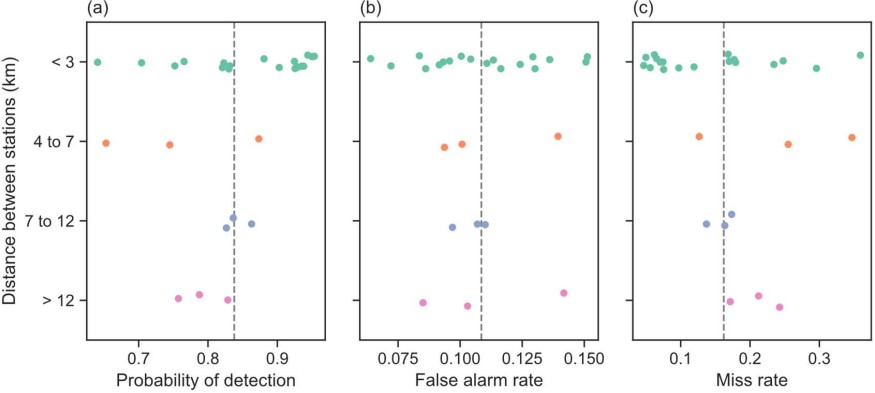


**Figure 13: Comparison of the (a) Probability of detection, (b) false alarm rate, and (c) miss rate according to the distance between the study site and paired weather station. The dashed grey line corresponds to the metric computed on the full dataset in Figure 12.**

## 5.3    Data validation across studies

The phase observations from this study can be compared to other studies that use different validation data, such as direct observations (e.g. Behrangi et al., 2018; Dai, 2008; Jennings et al., 2018). However, it can be difficult to compare the phase occurrence according to the 2-m air temperature, as the datasets in such studies often exclude mixed-phase precipitation. Consequently, the mixed phase is usually not analyzed in detail. One method to simply compare

phase partitioning models is the critical threshold air temperature value $CT_a$, which is defined as the critical temperature threshold where both solid and liquid phase have 50% chance of occurrence. In the case of this study, we define a different critical threshold for solid $CT_S$ and liquid phase $CT_L$, as well as a temperature where the probability for mixed phase is highest $P_m$. Figure 14 shows the probability of occurrence of the phases at the study sites separated in 0.2°C

bins. The resulting thresholds are $CT_S$ of 1.3°C, $CT_L$ of 3.8°C and $P_m$ of 2.4°C for a mixed-phase probability of 0.44. It is also noteworthy that $P_m$ is roughly where the probability of solid and liquid precipitation is equal. Because of this study's aggregation step, $CT_S$ should be similar to $CT_a$ values from other studies, as the aggregation mostly affected the probability of mixed and liquid precipitation, and $CT_L$ will be much warmer than $CT_a$.

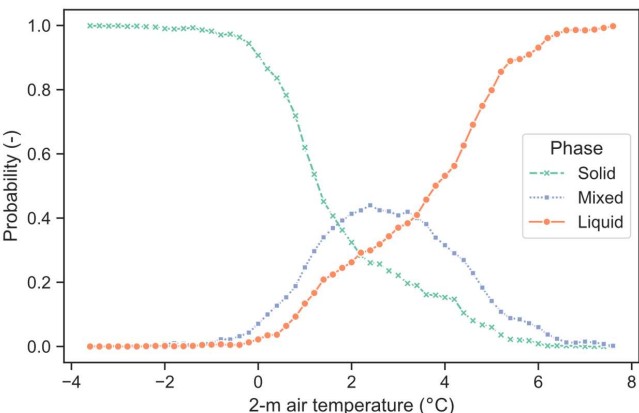


**Figure 14: Observed probability of occurrence for the solid, liquid, and mixed phase.**

In Behrangi et al. (2018), the average hourly $CT_a$ of 1.58°C aligns with the study's $CT_S$ and the
calibrated 2-m air temperature threshold of benchmark ST. One of the main conclusions from
the study was that the wet-bulb temperature model is more robust than the dry-bulb temperature
model, as the $CT_a$ can vary significantly from site to site. The study's $CT_a$ has an upper limit of
2.16°C, which closely matches the $P_m$ of this study. This finding lends credibility to the
disdrometer phase identifications and the phase aggregation step, as it indicates the temperature
range in which both solid and liquid phases are possible.

In Dai (2008), the overland 3-hourly $CT_a$ of 1.2°C is comparable to this study's $CT_S$, despite the
different time step. The chance for mixed phase in this study is much higher and more likely at
warmer temperatures than that of Dai (2008), where they report a peak 14.3% chance of mixed
rain and snow at 1.4°C overland.  However, Ding et al. (2014) have shown that the probability
of mixed-phase precipitation at the daily timestep greatly increases in humid conditions,
particularly near saturation. Such an analysis would, however, be required at the hourly time
step to confirm this behavior. The reasoning for the increase in mixed-phase precipitation
probability is that the increase in relative humidity decreases evaporative cooling and favors a
transition from snow to rain. In contrast, the temperature difference between the hydrometeors
and the air decreases as humidity rises, which decreases sensible heat transfer and hinders the
transition from snow to rain. The relatively homogenous conditions of the study sites could
explain the differences in mixed-phase precipitation probability, while the analysis in Dai (2008)
lumped together a large amount stations.

The findings in Jennings et al. (2018) report a much lower 3-hourly $CT_a$ of 0.7°C for
precipitation in 90-100% relative humidity and 0.9°C for precipitation occurring in 90-105 kPa,
humidity and pressure conditions where the majority of this study's precipitation occur. The
source of the validation data, direct manual observations versus automated observations, could


account for the difference. The longer time step may lead to a lower critical threshold because the energy needed to melt the precipitation can be supplied over a longer period.

Overall, the radar-based disdrometer measurements are similar to the findings of previous studies, but more research is needed to properly quantify the uncertainties associated with this
type of disdrometer. In addition, models based on automated phase observations may differ from those based on direct observations, especially as the time step can vary from study to study. This also highlights the importance of the verification step performed after aggregating mixed snow and rain/drizzle with rainfall, as their effect was deemed closer to that of rainfall.

## 6   Conclusion

The study used phase measurements from radar-based disdrometers to train probabilistic models to classify and partition precipitation data for a network of study sites in eastern Canada. The study sites were located in predominantly boreal climates and at similar elevations, ranging from 315 to 641 meters above sea level. The mean annual 2-meter air temperature was around 0.2°C, and the cumulative annual precipitation was significant at 902 mm. The humidity conditions for
the data points used in the study were generally close to water vapor saturation. The utilization of automated measurements enabled partitioning of precipitation for mixed-phase events, which were previously limited with direct phase observations. The studied PGP models showed an improvement in phase partitioning with prior phase classification compared to benchmark models of varying complexity. PGP provides more accurate phase classification, which can
benefit hydrological modeling at both local and watershed scales. It successfully reproduced the phase overlap between 1.5 and 3.5°C, where mixed phase probability was the highest.

The classification performances show a substantial enhancement in phase classification as opposed to benchmark models, which were designed to minimize errors in phase partitioning. Additionally, the PGP models reduced partitioning error, especially PGP_hydromet and
PGP_full. However, due to prior classification, partitioning performance is highly dependent on classification performance. As a result, the less complex PGP_basic had increased error variability. According to the input variable importance analysis, atmospheric pressure was the second most important hydrometeorological variable for phase classification. The reanalysis atmospheric data reduced the partitioning error variability of PGP_full in comparison to the other
PGP models. As for relative humidity, it was deemed to be the least important hydrometeorological feature for phase classification due to the regional homogeneity of the study sites. Overall, these findings demonstrate that automated phase observations enhance PGP method development and significantly improve precipitation phase classification, even with limited hydrometeorological information. The incorporation of reanalysis atmospheric data


further enhances the accuracy of local observations, pointing towards potential operational applications for such methods.

The presented methodology could be applied to other environments, including drier conditions or a broader spectrum of environments. Further research should include a comprehensive comparison of the radar-based disdrometers used in this study with other phase validation

techniques to assess potential limitations. Research is also needed to improve the prediction of the mixed phase. Other variables such as wind speed could be considered, as high wind speed can have a cooling effect on precipitation. Additionally, the impact of using a model that combines both phase classification and partitioning on snowpack accumulation and basin mass and energy dynamics should be investigated.





**Appendix A.    Study sites details**

**Table A1: List of study site coordinates, elevation, and timeframe where the station has been in operation.**

| Station name | Longitude (° W) | Latitude (° N) | Elevation (m ASL) | Operational timeframe |
|---|---|---|---|---|
| ARGENT | 69.778857 | 50.776574 | 641 | 2020-2023 |
| AUXLOUPS | 70.48743 | 51.90589 | 537 | 2020-2023 |
| BAUBERT | 63.55696 | 51.40946 | 541 | 2020-2022 |
| BETSIA_M | 69.913268 | 49.97657 | 403 | 2019-2023 |
| CABITUQG | 69.513611 | 49.573333 | 491 | 2019-2023 |
| CONRAD | 74.261131 | 47.607249 | 433 | 2020-2023 |
| DIAMAND | 73.18308 | 47.231302 | 373 | 2020-2023 |
| GAREMANG | 67.13986 | 51.11064 | 778 | 2020-2023 |
| HARTJ_G | 67.94598 | 51.77931 | 460 | 2020-2023 |
| LACROI_G | 70.07939 | 51.32871 | 621 | 2019-2023 |
| LAFLAM_G | 70.270496 | 48.930225 | 519 | 2019-2023 |
| LBARDO_G | 67.828433 | 51.111896 | 486 | 2019-2023 |
| LEVASSEU | 68.75496 | 51.26848 | 466 | 2020-2023 |
| LOUIS | 68.48974 | 49.88662 | 315 | 2020-2023 |
| LOUISE_G | 68.839767 | 50.658526 | 397 | 2019-2023 |
| MOUCHA_M | 69.52778 | 52.12551 | 565 | 2019-2022 |
| NOIRS | 68.83173 | 50.12116 | 385 | 2020-2023 |
| PARLEUR | 69.52237 | 51.28547 | 485 | 2020-2021 |
| PERDRIX | 67.96745 | 50.12946 | 315 | 2022-2023 |
| PIPMUA_G | 70.91581 | 49.36052 | 566 | 2020-2023 |
| PORTO | 70.27591 | 49.58423 | 413 | 2020-2023 |
| ROUSSY_G | 68.09436 | 50.42347 | 456 | 2020-2023 |
| RTOULNUS | 67.47676 | 50.96475 | 688 | 2020-2022 |
| SAUTEREL | 63.83804 | 51.91782 | 459 | 2020-2023 |
| STMARG_G | 67.04636 | 51.89198 | 461 | 2020-2023 |
| WABISTAN | 73.441157 | 48.484572 | 565 | 2020-2023 |





**Appendix B. Disdrometer phase identification validation**

Snow water equivalent (SWE) and snow depth observations were compiled from the entire network on a winter-by-winter basis. If more than 30% of a winter's snowpack observations were missing at a station, the winter is not included in this analysis. The resulting data subset consists of 11 winter-sites with a total of 53,520 hourly data points. The hourly data was then separated into precipitation events. The following filters were applied to the events:

- Duration ≥ 3 h,
- Mean 2-m air temperature between −5 and 5°C,
- Total precipitation ≥ 0.5 mm,
- Mean SWE ≥ 15 mm.

This filtering step aimed to exclude short events and events that occurred either in warmer
conditions, where phases other than rain are uncommon, or in the absence of snow cover. As such, 235 precipitation events were retained. In addition to the data points encompassing each event, the following hours were added until the next update of the SWE observations, at most 6 hours. The events were then classified according to their main precipitation phase, that is the phase associated with at least half the total precipitation of the event.

The mean 2-m air temperature, SWE variation (ΔSWE), and snow depth variation (ΔSD) are compiled from precipitation events, according to the main precipitation phase of the event (Table B1). The effects of rain and the mix of snow and rain/drizzle events on the snowpack are similar, a SWE increase accompanied by a SD decrease. In addition, the average temperature of mixed snow and rain/drizzle events is significantly above the freezing point, where rainfall is more
likely to occur than snowfall. In the case of freezing rain, the average temperature during the events is more similar to snowfall. Although freezing rain does not generally increase the SD, it contributes to the solid component of the snowpack as it freezes on contact. Thus, the phase aggregation of this study was based on the hydrological impact and temperature range of freezing rain and the mix of snow and rain/drizzle.

**Table B1: Precipitation events characteristics separated by phase.**

| Main phase | Event count | Mean temperature (°C) | ΔSWE (mm) | ΔSD (cm) |
|---|---|---|---|---|
| Snow | 192 | -2.0 | 3.0 | 3.3 |
| Rain | 12 | 3.6 | 3.5 | -1.8 |
| Mix of snow and rain/drizzle | 19 | 1.5 | 5.9 | -1.5 |
| Freezing rain | 12 | -1.7 | 3.2 | 0.2 |



**Appendix C.    Benchmark models description**

The single threshold model ST to compute the solid precipitation fraction $f_{snow}$ (-) functions as follows:

$$f_{snow} = \begin{cases} 1 & T_a \leq T_K \\ 0 & T_a > T_K \end{cases} \tag{C1}$$


Where $T_a$ is the temperature (°C) and $T_K$ is the calibrated temperature threshold (°C). The linear transition model LT uses two calibrated thresholds to calculate $f_{snow}$:

$$f_{snow} = \begin{cases} 1 & T_a \leq T_{snow} \\ \dfrac{T_{rain} - T_a}{T_{rain} - T_{snow}} & T_{snow} < T_a < T_{rain} \\ 0 & T_a \geq T_{rain} \end{cases} \tag{C2}$$

Where $T_{rain}$ and $T_{snow}$ are the calibrated rain and snow thresholds (°C). Finally, the psychrometric energy balance model PB (Harder and Pomeroy, 2013) calculates $f_{snow}$ as follows:

$$f_{snow} = \frac{1}{1+b+c^{T_i}} \tag{C3}$$

Where $b$ and $c$ are calibrated values and $T_i$ is the hydrometeor temperature (°C). $T_i$ is calculated iteratively with the following function:

$$T_i = T_a + L_t \frac{D}{\lambda_t}\left(\rho_{T_a} - \rho_{T_i}\right) \tag{C4}$$

Where $L_t$ is the latent heat of sublimation or vaporization (J k$^{-1}$), $D$ is the diffusivity of water
vapour (m$^2$ s$^{-1}$), $\lambda_t$ is the thermal conductivity of air (J m$^{-1}$ s$^{-1}$ K$^{-1}$) and $\rho_{T_a}$, $\rho_{T_i}$ are the water vapour density of the surrounding air and on the hydrometeor's surface respectively (kg m$^{-3}$). The procedure to compute the variables is as detailed in Harder and Pomeroy (2013). $D$ is computed following Thorpe and Mason (1966):

$$D = 2.06 \times 10^{-5} \left(\frac{T_a}{273.15}\right)^{1.75} \tag{C5}$$


The vapour pressure $e$ (kPa) is computed from Dingman (2015):

$$e = \frac{RH}{100} \times 0.611 \exp\left(\frac{17.37T}{237.3 + T}\right) \tag{C6}$$



Where $RH$ is the relative humidity (%) and $T$ is the air temperature (°C). $\rho$ is computed following the ideal gas law:

$$\rho = \frac{m_w e}{RT} \tag{C7}$$

Where $m_w$ is the molecular weight of water (0.01801528 kg mol$^{-1}$) and $R$ is the universal gas constant (8.31441 J mol$^{-1}$ K$^{-1}$). The air thermal conductivity $\lambda_t$ is computed from List (1951):

$$\lambda_t = 0.000063 T_a + 0.00673 \tag{C8}$$


Finally, the latent heat of sublimation ($T_a < 0$) and vaporization ($T_a \geq 0$) are computed as follows (Yau and Rogers, 1996):

$$L_t = \begin{cases} 1000(2834.1 - 0.29 T_a - 0.004 T_a^2) & T_a < 0 \\ 1000(2501 - 2.361 T_a) & T_a \geq 0 \end{cases} \tag{C9}$$

Table C1 shows the calibrated parameters for the models presented in this section. The calibration was made on the same training set used for the PGP models. Figure C1 shows the simulated solid fraction for the benchmark models, as well as the observed solid fraction.

**Table C1: Benchmark model calibrated parameters.**

| Model | Calibrated parameters |
|---|---|
| Single threshold (ST) | $T_K = 1.50$ |
| Linear transition (LT) | $T_{snow} = -0.38$ |
| | $T_{rain} = 5.00$ |
| Psychrometric balance (PB) | $b = 6.34$ |
| | $c = 0.39$ |

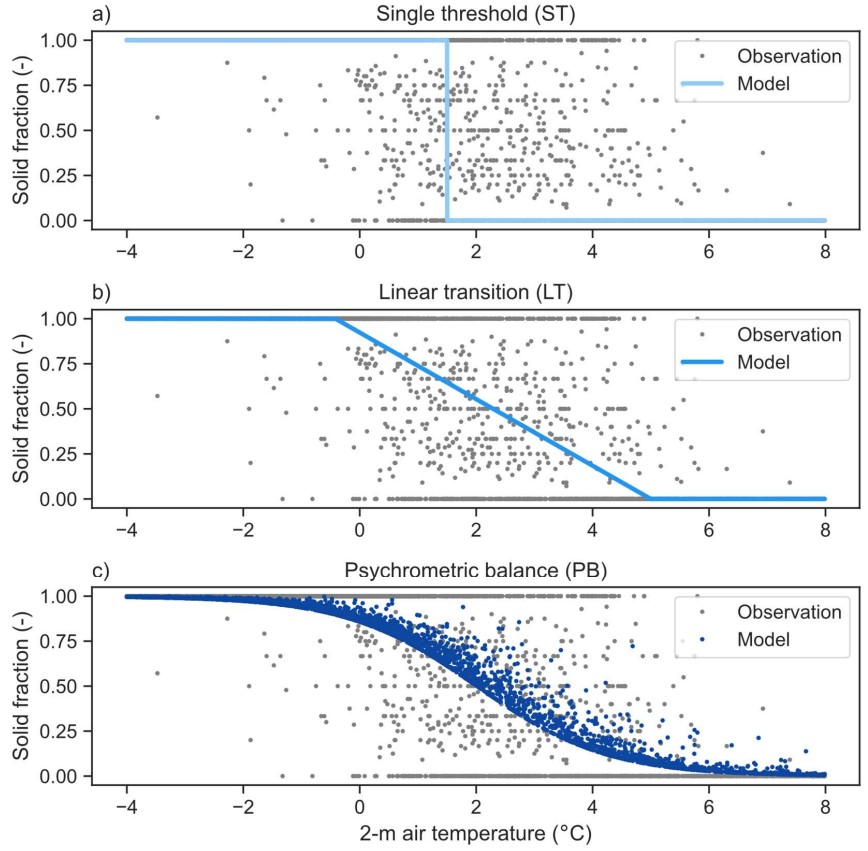


**Figure C1: Observed solid precipitation fraction according to the 2-m air temperature and the modeled solid precipitation fraction of the a) static threshold, b) linear transition and c) psychrometric balance.**



## 7    Data availability

The data used to train/calibrate the models in this study are available at 10.5281/zenodo.10790810. Supplementary data in the analysis are available from the corresponding author upon reasonable request. All data are subject to Hydro-Québec's Creative Commons Attribution – Non-Commercial 4.0 International licence (https://www.hydroquebec.com/documents-data/open-data/licence.html).

## 8    Author contributions


ABT, DFN, and FA designed research. FP and AV provided the study site data, as well as technical guidance. ABT with the help of OC performed data clean-up and corrections. ABT analyzed data and devised the precipitation partitioning models with inputs from DFN, FA, and JMT. ABT wrote the paper with inputs from DFN, FA, and JMT, as well as comments from OC,
FP, and AV.

## 9    Competing interests

The authors declare that they have no conflict of interest.

## 10    Acknowledgements

This project was financially supported by the Quebec Ministry of Public Safety and by the
Natural Sciences and Engineering Research Council of Canada through the Alliance program (grant ALLRP 549108 – 19 entitled 'Climate projection for hydrological applications in cold regions region, EVAP-2) with Hydro-Québec and Ouranos as industrial partners. We also thank M. Bédard-Therrien for their guidance in the development of machine learning models.

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
