# Peer review of "Leveraging a radar-based disdrometer network to develop a probabilistic precipitation phase model in eastern Canada"

_Hydrology and Earth System Sciences, 2024_

## Author Comment (AC1)

**HESS-2024-78 Leveraging a Disdrometer Network to Develop a Probabilistic Precipitation Phase Model in Eastern Canada**

**Response to anonymous referee 1**

We would like to thank the reviewer for their helpful contribution to the article. Please note that additions to the article are shown in bold. The lines in this document refer to the previous version of the manuscript and may be subject to change in the revised version.

**1    General comments**

This study of predicting the phase of precipitation is rather regional, but it is worthwhile for the large amount of novel data used and different conclusions on the importance of humidity to earlier studies in other regions. My comments are minor or technical.

Thank you, your kind words are greatly appreciated.

**2    Specific comments**

2.1    Considering that there are several types of disdrometer, it might be worth getting the word "radar" into the title.

This is a good point, thank you. We propose the new title:

Lines 1-3: *Leveraging a **radar-based** disdrometer network to develop a probabilistic precipitation phase model in eastern Canada*

2.2    Line 21: Specify which reanalysis data are in PGP_full (thickness, temperature lapse rate and surface pressure).

Thank you, we agree that it would be beneficial to detail which variables are used for each model. We suggest to the following change:

Lines 20-21: *PGP_basic is based on 2-m air temperature and site elevation, while PGP_hydromet integrates relative humidity, **surface pressure and precipitation rate**. The PGP_full dataset **includes all the previous data, along with additional reanalysis data, that is, 1000-850 hPa layer thickness, and temperature lapse rate.***

2.3    Line 71: The difference in computational resources between threshold and curvilinear functions is trivial in application.

This is a fair point, we suggest removing the corresponding part of sentence, resulting in the following:

 *Furthermore, these types of methods can be refined into curvilinear functions, which would theoretically yield to a more accurate phase identification (Feiccabrino et al., 2013).*

2.4    Line 189: The WMO definition of freezing rain is supercooled liquid drops that freeze on impact with the ground. Is this what the WS100 records as "freezing rain"? How does its diameter-fall velocity relationship differ from warm rain?

Regarding the instrument's definition of freezing rain, considering that the instrument outputs files that use WMO codes and that freezing rain is the term used in official documentation, we assume it corresponds to the WMO definition.

As for the diameter-fall velocity relationship of freezing rain, this is a good point and would be pertinent information to add. Unfortunately, after verification with the company manufacturing the disdrometer (Ott HydroMet), the specific details of this relationship are proprietary information that they keep confidential at his time.

2.5    Figure 2: Picking up some hints in the text, would scatter plots of these variables be interesting?

There is not enough data and spatial variability to truly have an interesting trend for scatter plots. However, we suggest adding a color scale for different latitudinal ranges, to supplement the figure interpretation. Here is the modified figure and the corresponding caption:

[Figure]

Lines 199-200: *Figure 2: Distributions of the (a) annual mean temperature, (b) annual mean precipitation and (c) elevation at the study sites, **separated by latitudinal range.***

Additionally, we propose modifying the following line to better fit the information in the figure:

Lines 175-176: *The southernmost sites (<**49**°N) receive **more** precipitation **on average**, with an annual mean of 1002 mm. **However, the variability observed at these sites is much greater than that observed elsewhere.***

**2.6 Line 240: Thickness does not indicate the travel time unless the hydrometeor fall velocity is also known.**

This is a good point; we propose removing the corresponding part of the sentence:

Lines 239-242: *The layer thickness between the two pressure levels is correlated with the mean temperature of the **and** is also commonly used in operational meteorological models (Feiccabrino et al., 2015).*

**2.7 Line 264: Aggregation of freezing rain with snow rather than rain is common in previous studies and is justified by the hydrological influence, but it seems to be a misclassification of the phase of the hydrometeors.**

We acknowledge that this aggregation could constitute a misclassification of the phase. However, the goal of this study is to provide a phase model for hydrological purposes, where freezing rain is generally not considered. For example, SWAT (Arnold et al., 2012) is widely used and lumps freezing rain with snow for snow accumulation. Thus, the important distinction for such a model is whether the precipitation increases the solid or liquid content of the snowpack, without regard if it contributes as fresh snow or an ice layer.

Furthermore, as it is by far the rarest precipitation phase, we judged that this aggregation would not impact significantly the results. The instances where freezing rain occurred consist of 6.4% and 3.6% of the solid and mixed phase precipitation data points, respectively. Freezing rain accounts for approximately 3% of the total winterly precipitation amounts. While this aggregation could affect the prediction of mixed-phase events that include freezing rain, it is such a small portion of the dataset that the effect is likely to be minimal.

**2.8 Figure 3: After reading the text many times, I think that the "mix of snow and rain/drizzle" in Figure 3a combines what the disdrometers class as "mix of snow and rain/drizzle" (which gets aggregated with the liquid phase in 3b) and hours with 15-minute periods classed as both snow and rain (which remain classed as "mixed" in 3b). But I am not confident in that interpretation (and I have no idea why the peak in snowfall close to 0C**

appears to go down slightly when classed as solid precipitation). Please don't make the reader work so hard on something simple.

The information conveyed in Figure 3 should indeed be made clearer, thank you for pointing this out. We propose the following modifications to the figure:
1. The upper part shows the 15-min precipitation data points that are used in the aggregation along an interpolated 15-min 2-m air temperature.
2. Adding "15-min" and "Hourly" to the count labels to clearly indicate the difference between Figures 3a and 3b.

[Figure]

Lines 278-279: Figure 3: *Event* counts of (a) the *15-min* precipitation phase identified by the disdrometers and (b) the aggregated *hourly* precipitation phases according to the 2-m air temperature.

We also propose the following modifications to the text supporting the figure:

Lines 280-280: *Figure 3a **shows the phase occurrence of the coupled 15-min precipitation data along an interpolated 15-min 2-m air temperature. The phase occurrence in Figure 3b** shows that **it is mostly the** mixed and liquid precipitation **distributions that are affected by the aggregation**, and that the very few freezing rain events that are aggregated with snow events result in the snow and solid phase distributions being very similar.*

**2.9    Figure 9: The colours do not convey any information, so I would not use them.**

While this is true in the case of this figure, we deemed it important to keep the colour scheme consistent between Figures 8 and 9 to quickly identify the different models.

**2.10   Line 572: What is "Improving PGP models' ability to accurately predict the mixed phase is manifold" meant to mean?**

This sentence is indeed rather vague and superfluous. We suggest rewording the paragraph:

Lines 571-575: ***The scoring scheme for permutation importance must be carefully selected according to the model and the use case. In this instance,*** *the PGP models tend to overpredict the mixed phase, which also negatively impact their ability to predict the other phases. In turn, this also affects the models' partitioning error,* ***which indicates that their overall performance is reliant on accurate phase classification****. For these reasons, the chosen scoring scheme for the permutation importance is the weighted F1 score, to consider* ***the classification of*** *the imbalanced* ***phase dataset****.*

**2.11   Figure 10: The diagonal is redundant. Removing it would allow making the rather small labels a bit bigger. The colour scale should have a label.**

Both points are valid. Including the diagonal does not add information and will be removed. Regarding the color scale, it is true that the reader should be able to understand that it indicates the correlation between variables without reading supporting text, so it will be added.

[Figure]

Figure 10: Correlation matrix of PGP_full input variables pairs.

**2.12 Line 825: Why exclude snowfall when there is not already snow on the ground?**

This is a good point, and the reasoning was not well explained here. The criterion was introduced to retain events where the SWE was greater than the instrument's lower measurement limit of 15 mm. We suggest the following changes:

Lines 824-825: *This filtering step aimed to exclude short events and events that occurred either in warmer conditions, where phases other than rain are uncommon, or in the absence of a snow cover **detectable by the instrumentation in place***.

**2.13 Appendix C: Does equation C4 not give the temperature for unventilated hydrometeors? The ice bulb temperature may be a more appropriate predictor (or there may be little difference due to the high relative humidities in this study).**

It is indeed the equation for the temperature for unventilated hydrometeors. The description of the equation could benefit from a little more detail. We suggest adding the following:

Lines 855-866: ***Based on the mass balance of a sublimating ice sphere, the temperature of an unventilated hydrometeor*** $T_i$ *is calculated iteratively with the following function:*

Ice bulb temperature may indeed be an appropriate predictor for drier climates. Among other predictors, the wet bulb (rather than ice bulb) temperature was tested in this study, as multiple studies have successfully done so. However, as hypothesized by the reviewer, the high relative humidities meant that there was very little gain in performance.

**3    Technical comments**

**3.1    Line 74: "yield a more accurate"**

Thank you, this will be corrected.

**3.2    Line 86-89: "it" becomes "they" over the course of this sentence.**

Thank you, "they" will be replaced by "it".

**3.3    Line 107: "Therefore" seems incorrect at the start of this sentence.**

This is true, it will be corrected.

**3.4    Line 168: "elevations range"**

Thank you, it will be corrected.

**3.5    Line 207: "with sensors"**

Thank you, it will be corrected.

**3.6    Line 208: Delete "ground".**

We assume that this comment refers to the unnecessary addition of level in "above ground level" and will correct it.

**3.7    Line 238 (and subsequently): Prevent automatic capitalization of the first word after a display equation when there is not a new sentence.**

Thank you for pointing this out, it will be corrected.

**3.8    Line 314: "harshly penalizes a poor score in either" has already been said.**

This part of the sentence will be removed, as it is indeed redundant.

**3.9 Line 316: "the model partitioning performances are"**

Thank you, this typo will be removed.

**3.10 Line 359: "the predicted phase is either solid or liquid,"**

Thank you, it will be corrected.

**3.11 Line 453: "PGP_basic overpredicts the mixed phase"**

Thank you, the sentence will be changed.

**3.12 Line 512-513: Spurious line break**

Thank you, it will be removed.

**3.13 Line 658: "importance of 1000-850 hPa layer thickness"**

The sentence will be modified accordingly.

**3.14 Line 679: "such as laser disdrometers"**

Thank you, this will be corrected.

**3.15 Line 686: "at a site sheltered from the wind"**

This will be corrected, thank you.

**4  References in this document**

Arnold, J. G., Moriasi, D. N., Gassman, P. W., Abbaspour, K. C., White, M. J., Srinivasan, R., Santhi, C., Harmel, R. D., van Griensven, A., Van Liew, M. W., Kannan, N., and Jha, M. K.: SWAT: Model Use, Calibration, and Validation, Transactions of the ASABE, 55, 1491-1508, https://doi.org/10.13031/2013.42256, 2012.

---

## Author Comment (AC2)

**HESS-2024-78 Leveraging a Disdrometer Network to Develop a Probabilistic Precipitation Phase Model in Eastern Canada**

**Response to anonymous referee 2**

Many thanks to the reviewer for the very insightful comments, which helped us improve the quality of the article. Please note that additions to the article are shown in bold. The lines in this document refer to the previous version of the manuscript and may be subject to change in the revised version.

**1   General comments**

This is a very nice paper, and it is written well. It includes a lot of detailed analyses and discussions that make the paper very informative. I think it fits the journal well. I recommend major revision as I have several minor comments that need to be addressed before I can accept the paper for publication. Otherwise, the paper is in good shape. Also, as you revise the paper, please make sure to clarify how the automatic measurements enable mixed phase classification/ partitioning?

Thank you for the kind words. Regarding how automatic measurements enable mixed phase classification/partitioning, we explain in lines 112-136 how automatic phase measurements allow mixed-phase partitioning. We propose adding the following sentence clarify the reasoning:

Lines 122-123: *High-frequency automatic measurements do not suffer from limitations caused by mixed-phase precipitation (Froidurot et al., 2014; Harpold et al., 2017)**, as the precipitation phase can be coupled with a concurrent precipitation amount. When both phase identification and precipitation gauge measurements are made at a high frequency, phase-separated precipitation can be compiled for hourly or more timesteps, thus allowing for mixed-phase partitioning.***

Also, I suggest adding "radar-based" in the title, to clarify this is not for laser-based dendrometer.

This comment is in line with Reviewer #1 (see comment 2.1). We agree that this distinction with laser-based disdrometers is important. We suggest the new title:

Lines 1-3: *Leveraging a **radar-based** disdrometer network to develop a probabilistic precipitation phase model in eastern Canada*

**2 Specific comments**

**2.1 Lines 21, list the atmospheric variables used.**

This was also mentioned by Reviewer #1. We agree with the detailing of the variables used. Please refer to the response under their comment 2.2 regarding the proposed changes.

**2.2 Abstract: partitioning is not an obvious term. I think by partitioning you mean the amount of each phase versus the type (classification). It would be helpful to define partitioning upfront.**

This is indeed the intended difference between the two terms, and we acknowledge that it is not a well-established distinction. We propose adding a sentence that makes a clearer distinction between the two:

Lines 16-17: *Single-phase precipitation was also found to occur more frequently than mixed-phase precipitation.* ***This outlines the need to classify the precipitation phase, as well partitioning correctly between the solid and liquid precipitation amounts in the case of the mixed phase.***

**2.3 Line 65, you may want to add one or both of these also to further support your point: https://doi.org/10.1016/j.jhydrol.2022.127884; https://doi.org/10.1029/2019GL084221**

This is greatly appreciated input. The proposed references will be added to the following part of the manuscript:

Lines 63-66: *Indeed, solid precipitation is much more sensitive to undercatch (underestimation due to the wind moving hydrometeors away from the gauge) than liquid precipitation (Rasmussen et al., 2012). Consequently, an inaccurate measurement of the phase necessarily translates into an erroneous estimation of the precipitation quantity.* ***Ehsani and Behrangi (2022) showed that undercatch for solid precipitation introduced a significant bias in gridded precipitation products at both the seasonal and annual scales at higher latitudes. This highlights the need to account for the precipitation phase at the synoptic scale, especially when using precipitation products to bias-correct satellite precipitation estimates (Behrangi et al., 2019; Ehsani and Behrangi, 2022).***

**2.4 Line 145, I thought the use of "However" instead of "Additional" might fit the sentence better.**

We agree that it is a better fit for the sentence, it will be changed.

**2.5** Line 153, I thought it would be useful to discuss the difference between phase classification and partitioning here as this may not be obvious for readers.

This is a good point, as it would further reinforce the distinction between the two terms. We propose adding the following sentence:

Lines 152-154: *The precipitation phase is classified before partitioning to accurately replicate its intricate behavior and to take advantage of the significant amount of validation data available through such a network.* ***As such, the models classify the precipitation as either solid, liquid, or mixed phase. The predicted phase then dictates the partitioning into solid and liquid fractions.***

**2.6** 185-186, so my understanding is that disdrometer does not "observe" phase. As you said in line 190 precipitation phase is identified according to the hydrometeor diameter-fall velocity relationships for water droplets and solid particles. So maybe you should replace "observation" with "estimation" or something similar.

This interpretation is correct, and it would indeed be more accurate to use terms such as "estimation" or "identification". We propose using the term "identification" in the paragraph from lines 185-197 containing the concerned sentence. However, for simplicity, we propose referring to the phase identifications as observations for the rest of the manuscript and adding at the end the following disclaimer.

Line 195: ***For simplicity, the phase identifications derived from the diameter-fall velocity relationships are referred as observations in this study.***

**2.7** Lines 280-285, it is not clear to me how the aggregation of the mix of snow and rain/drizzle with rain is performed? How did you decide to convert fractions to solid or liquid phase?

The official WMO description of the mix of snow and rain/drizzle is "Rain or drizzle and snow, moderate or heavy", and under the broader rainfall category. This fact, and the verifications that are shown in Appendix B, suggest that this type of precipitation behaves more like rainfall than snowfall. Thus, for the purposes of this study, we assume that most of the precipitation is therefore rain accompanied by near-melting snow, resulting in a fully liquid precipitation. Additionally, the wording of the phase name was chosen purely for readability. As such, we also propose changing the name to "mix of rain or drizzle and snow" to better put an emphasis on the rain/drizzle part of the precipitation phase.

We acknowledge that, given the wording of the precipitation phase, the partitioning could be assumed to be 50/50 for the solid and liquid precipitation fractions, as seen for example in Wayand et al. (2016). This assumption was tested with this the data from this study and yielded unexpected results, such as a 2-m air temperature critical threshold for solid precipitation of about 3°C. This seemed highly unlikely given the literature on this subject (e.g. Jennings et al.,

2018). Additionally, the discussion comparing the data used in other studies (see section 5.3) lends credibility to this study's aggregation step, as the results are similar.

We propose indicating this study's assumptions made about the fraction of solid and liquid precipitation in the mix of snow and rain/drizzle in a clearer way:

Lines 262-264: *Therefore, the disdrometer identifications of freezing rain and of mix of snow and rain/drizzle were aggregated,* **respectively,** *with snow and rain events.* ***For example, if an hourly precipitation has a fraction of rain and a mix of rain/drizzle and snow, it would be considered completely liquid after the aggregation.***

Furthermore, we propose adding the following statement to the section discussing instrument accuracy.

Lines 675-677: *The phase aggregation step considered the behavior of the snowpack following the different phases detected by the disdrometers.* ***Following a mix of rain/drizzle and snow, the SWE and snow height tended to decrease, a snowpack response similar to that following a rain event. It can be inferred that this type of precipitation is likely to be dominated by rainfall given the warm temperature at which it occurs and the ensuing effects on the snowpack. However, it is probable that this interpretation is specific to the disdrometers used in this study, unless evidence to the contrary emerges.*** *Phase identification errors also have the potential to introduce uncertainties in the results****, notably in the case of mixed-phase precipitation****.*

**2.8 Add reference for the performance metrics used, on the other hand POD, FAR, HSS, CSI, etc. are also very popular. I also like BIAS as it shows over or under detection. How can you say you over or underpredicted?**

The performance metrics used in this study are common in the machine learning field, a reference from Rokach et al. (2023) will be added. Recall is the term for POD in machine learning applications, thus a short acknowledgment of this fact will be added to the paper. As such, the precision was a logical choice for a second performance metric, for its inverse relationship with recall. The precision is also related to FAR, as it indicates the ratio of accurate predictions rather than false predictions. HSS and CSI were also calculated for this study but were not presented as they did not give significantly different results than the presented F1 score.

The assessment of over and underprediction can be achieved with precision and recall. Low precision and high recall scores indicate an overprediction (i.e. the mixed phase prediction of the linear transition and psychrometric balance models). On the contrary, high precision and low recall indicates underprediction. This interpretation will be added to the description of the metrics:

Lines 295-299: *The combination of precision and recall is commonly used to evaluate model classification performance, as the metrics indicate different information. Precision indicates the proportion of correct predictions for a given phase, while recall indicates the **probability of detection** for a given phase. **By definition**, model precision and recall are inversely proportional. **The assessment of both metrics informs if a model over or underpredicts a given class. For instance, low precision and high recall indicate a class overprediction, while high precision and low recall indicate a class underprediction.** Therefore, a model that achieves good performance in both metrics is desirable.*

2.9    Line 316, don't you have an extra "model" in the sentence?

This is indeed the case; it will be corrected.

2.10   L323-324: I am not sure if I understand why RMSE is the same for liquid and solid phase. Explain.

In the case of this study, the RMSE on either the solid or liquid precipitation is equivalent to the root mean squared misclassified precipitation amounts. As such, if the error of predicted solid precipitation is of X mm, the error on liquid precipitation is -X mm. Of course, because of the squaring of the error, both RMSE are equal. Thus, we propose replacing the following lines to better present this reasoning:

Lines 323-325: ***Because of the partitioning between solid or liquid precipitation, the RMSE is equal to the root mean squared of the misclassified precipitation. Therefore, the RMSE is the same for both solid and liquid precipitation, and a single score is presented.***

2.11   Section 3.3. it is not clear. Did you use the same number of solid and liquid phase for training? How about testing? For example, you say 60% solid and 26% liquid were there. Did you reduce the number of solid in training to match the number of liquid phase samples?

We acknowledge that the method used was not clearly explained. The stratified K-fold method was used to maintain the precipitation phase proportions of the dataset for both the training and validation sets, to ensure that there were enough liquid and mixed phase samples in the subsets. We propose the following additions to the manuscript:

Lines 351-352: *The data were then split using an 80/20 ratio between the training and validation sets respectively, resulting in 13,339 data points for training and 3,335 for validation. **To account for the prevalence of solid precipitation samples**, the training and validation sets were stratified to maintain the **aforementioned** phase proportions between the two subsets **(60% solid, 26% liquid, 14% mixed)**.*

**2.12** Section 3.5. line 390, in the PB method, is the hydromet temperature similar to Wet bulb temperature. If not, what's the difference between PB and wet bulb temperature?

While the wet bulb temperature is based on the vapor deficit of an air parcel, the hydrometeor temperature is based on the mass and energy balance of a sublimating ice sphere. In practice, they are most likely similar, since the hydrometeor's surface is considered saturated with water vapor for mass transfer calculations. However, as the conditions are mostly near saturation in this study, both the hydrometeor and the wet bulb temperature are close to the dry bulb temperature. We propose adding more context to the manuscript:

Lines 389-391: *Finally, the psychrometric energy balance (PB) model is used, which is a phase partitioning method **based** on **the mass and energy balance of a sublimating ice sphere** that integrates the relative humidity to estimate the hydrometeor temperature (Harder and Pomeroy, 2013).*

**2.13** Line 395, not sure what you mean the simple threshold method also gives binary and there is no mixed phase. The probability approach can be converted to binary rain/snow. I am not sure if I understand your point.

The simple threshold model was included solely because it is still used in hydrological applications and is the simplest method available. Probabilistic models (e.g. Behrangi et al., 2018; Jennings et al., 2018) were not included as benchmark models because mixed-phase precipitation was omitted from these studies, resulting from the limitations of the direct phase observations. Direct phase observations being a qualitative measure, there is no way to properly partition the precipitation phase. Other studies using probabilistic models do include mixed-phase precipitation (e.g. Ding et al., 2014; Shin et al., 2022), however the same partitioning issue remains. We propose modifying the following parts to better explain the reasoning:

Lines 396-398: ***Previous studies that employed probabilistic models based on direct phase observations (e.g., Behrangi et al., 2018; Jennings et al., 2018) were not included as benchmark models. Mixed-phase precipitation is typically excluded from such studies, as there is no effective method to accurately partition the precipitation due to the categorical nature of direct phase observations. The above considerations make such models difficult to compare with the PGP models presented in this study.***

**2.14** Line 421, from Fig 6 I don't see how median RH is around 97%. Should be lower.

Please note that the vertical axis in Figure 6b) is on a logarithmic scale, therefore most of the data points are near saturation. To provide further context, roughly 65% of the data points are above 95% relative humidity, and 21% are at saturation. The use of a logarithmic scale was deemed necessary to improve readability of the figure, as seen in figure R1 which displays Figure 6b) without the logarithmic scale.

[Figure]

Figure R1: Relative humidity counts separated by precipitation phase.

**2.15 Lines 445-455: I don't see any evidence in supporting your performance evaluation. Do you have a graphic or statistics in support of what you stated here about the performance evaluation? If so, refer to them in your text.**

This is a very good point, as it is true that this statement has no supporting evidence at this point in the manuscript. We propose rearranging and merging the paragraphs from lines 445-455 and from lines 461-472.

Lines 445-455 and 461-472:

*Figure 7 shows the phase density distribution of the benchmark models and the PGP models. The **corresponding weighted** classification scores of the models **are presented** in Table 3. **The phase density distributions show the limitations of the benchmark phase partitioning models, namely that the mixed phase is absent or overrepresented compared to the observations. However,** ST performs well in all three metrics due to the low likelihood of mixed phase occurrence. When evaluating the overall classification performance using the F1 score, LT follows ST because of a disparity between precision and recall that affects its F1 score. The lower recall score for LT can be attributed to its overprediction of the less frequent mixed phase, which, in turn, negatively affects the recall of other phases. This enhances the model's weighted precision by decreasing the number of false positives in non-mixed-phase prediction. The same reasoning can be more extensively applied to PB's weighted scores. The mixed phase's overlap with other phases significantly decreases the model's overall recall. **Due to the relationships used to create the benchmark models, the overlap between all three phases is not accurately represented. By including relative humidity, PB can model the phase overlap, but this does not improve the modelled phase distribution densities with respect to the observations.***

*The weighted F1 score for the PGP models shows that they have a more robust performance, as they yield high weighted precision and recall scores, while having a small disparity between*

**both scores.** *The PGP models reproduce the observed phase overlap well, but slightly overpredict the mixed phase, affecting both the solid and liquid-phase predictions. PGP_basic overpredicts the most the mixed phase, while the difference between PGP_hydromet and PGP_full is marginal. This result suggests possible improvements to PGP models, particularly for mixed-phase precipitation.*

**2.16  Can you show the observation reference in Fig. 7?**

This is a good suggestion to improve the ease of comparison with the observations. Here is the modified figure that will be added to the manuscript.

[Figure]

*Figure 7: **Hourly** phase distributions according to 2-m temperature of the (a) **observations, (b)** single threshold, **(c)** PGP_basic, **(d)** linear transition, **(e)** PGP_hydromet, **(f)** psychometric balance and **(g)** PGP_full. PGP model details are summarized in Table 2.*

**2.17  Line 512, add "S" . Table 4 show"s"**

Thank you, this will be corrected.

**2.18 REMOVE lines 512-521. This is a repeat ! later in Lines 530-538:**

Thank you, this will be corrected.

**2.19 Line 571, I don't understand this sentence "Improving PGP models' ability to accurately predict the mixed phase is manifold"**

Reviewer #1 made a similar comment about this sentence as well. We agree that it is vague and superfluous. Here are the proposed changes:

Lines 571-575: *The scoring scheme for permutation importance must be carefully selected according to the model and use case. In this instance, the PGP models tend to overpredict the mixed phase, which also negatively impact their ability to predict the other phases. In turn, this also affects the models' partitioning error, which indicates that their overall performance is reliant on accurate phase classification. For these reasons, the chosen scoring scheme for the permutation importance is the weighted F1 score, to consider the classification of the imbalanced phase dataset.*

**2.20 Lines 627-628: Do you have references to back up your statement here?**

The statement is based on the results presented in this study, where both LT and PB greatly overpredicted mixed-phase precipitation. However, we infer that this is due to the calibration method. For instance, the mixed-phase prediction of the benchmark models could be artificially constrained to reduce overprediction and improve classification performance. Such constraints would impact the models' partitioning performance, as the benchmark model results were optimized for partitioning, hence the use of "trade-off" in the manuscript. We propose modifying the following to better reflect this reasoning:

Line 627-630: *These models were calibrated to minimize partitioning error, but in doing so, they are biased toward predicting mixed-phase precipitation. The mixed-phase prediction of the benchmark models could be artificially constrained to reduce overprediction and improve classification performance. Such constraints would however increase the benchmark models' partitioning error, given that they were calibrated according to solid precipitation fraction. Therefore, there is a trade-off between classification and partitioning error for precipitation fraction-based models such as LT and PB.*

**2.21 Line 635, could be helpful if you refer to the figure or table that supports your overprediction claim**

This is a good point, it will be added.

2.22  Line 785, can you remind based on which figure the "phase overlap between 1.5 and 3.5°C" was concluded?

This is a great point. To improve the overall coherence of the manuscript, it will be added. Here is the modified sentence:

Line 785-786: *It successfully reproduced the phase overlap between 1.5 and 3.5°C **seen in Figures 6 and 7**, where the probability of mixed phase was highest.*

**3  References to be added to the manuscript**

Behrangi, A., Singh, A., Song, Y., and Panahi, M.: Assessing Gauge Undercatch Correction in Arctic Basins in Light of GRACE Observations, Geophysical Research Letters, 46, 11358-11366, https://doi.org/10.1029/2019GL084221, 2019.

Ehsani, M. R. and Behrangi, A.: A comparison of correction factors for the systematic gauge-measurement errors to improve the global land precipitation estimate, Journal of Hydrology, 610, 127884, https://doi.org/10.1016/j.jhydrol.2022.127884, 2022.

Rokach, L., Maimon, O., and Shmueli, E.: Machine Learning for Data Science Handbook, 3, Springer Cham, https://doi.org/10.1007/978-3-031-24628-9, 2023.

**4  References in this document**

Behrangi, A., Yin, X., Rajagopal, S., Stampoulis, D., and Ye, H.: On distinguishing snowfall from rainfall using near-surface atmospheric information: Comparative analysis, uncertainties and hydrologic importance, Quarterly Journal of the Royal Meteorological Society, 144, 89-102, https://doi.org/10.1002/qj.3240, 2018.

Ding, B., Yang, K., Qin, J., Wang, L., Chen, Y., and He, X.: The dependence of precipitation types on surface elevation and meteorological conditions and its parameterization, Journal of Hydrology, 513, 154-163, https://doi.org/10.1016/j.jhydrol.2014.03.038, 2014.

Jennings, K. S., Winchell, T. S., Livneh, B., and Molotch, N. P.: Spatial variation of the rain-snow temperature threshold across the Northern Hemisphere, Nat Commun, 9, 1148, https://doi.org/10.1038/s41467-018-03629-7, 2018.

Shin, K., Kim, K., Song, J. J., and Lee, G.: Classification of Precipitation Types Based on Machine Learning Using Dual-Polarization Radar Measurements and Thermodynamic Fields, Remote Sensing, 14, 3820, https://doi.org/10.3390/rs14153820, 2022.

Wayand, N. E., Stimberis, J., Zagrodnik, J. P., Mass, C. F., and Lundquist, J. D.: Improving simulations of precipitation phase and snowpack at a site subject to cold air intrusions: Snoqualmie Pass, WA, Journal of Geophysical Research: Atmospheres, 121, 9929-9942, https://doi.org/10.1002/2016JD025387, 2016.

---

## Author Comment (AC3)

**HESS-2024-78 Leveraging a Disdrometer Network to Develop a Probabilistic Precipitation Phase Model in Eastern Canada**

**Response to Dr. James Feiccabrino**

We are grateful to Dr. Feiccabrino for his valuable input, which has clearly contributed to the improvement of our manuscript. Please note that additions to the article are shown in bold. The lines in this document refer to the previous version of the manuscript and may be subject to change in the revised version.

**1   General comments**

My recommendation (old standard) would be accept with minor corrections.

I overall enjoyed reading through the article and commend the writers for a well written thorough overview of the state of precipitation phase determination in hydrological or hydro-met models.  They did a very good job of stating the problem (why it's important, and reviewing past work on the issue).  They have an interesting and well thought out method and do a good job explaining their results and how it fits into the current state of work on the issue.  They also attempt to identify why some results don't agree with all previous work and how it could be used for future studies.  I would consider this a solid write-up.

I really liked how you explained the difference in outcomes between your study and other studies citing RH as an important factor for precipitation phase determination lines 605 - 609, and 641-644.  I wish more papers included notes like this.

Thanks for the positive feedback, it is much appreciated.

**2   Major corrections**

2.1   Lines 530-539 are an exact copy of lines 513-521.  One of these paragraphs should be deleted, and may affect the final location of Table 4.

Thank you, the paragraph on lines 513-521 will be removed, and Table 4 will be left as is.

2.2   Line 254 - (ECCC, 2024) is the reference used, but does not appear in that form in the reference list.  I believe this is the reference on line 932, but there is no easy way to link the reference in the article to the reference named "climate glossary".

This is a good point. After consideration, we propose replacing the reference with the relevant section in the WMO guide to instruments and methods of observation (WMO, 2018), as it is a more general reference than the ECCC climate glossary.

Lines 249-253: *A first filter was applied, where hourly precipitation rates < 0.2 mm h$^{-1}$ were considered erroneous trace amounts,* **following the standard WMO methodology (WMO, 2018)**.

Please see the section at the end of this document for the full reference.

**2.3** Line 225 - Hersbach 2023 is missing et al., in the article reference.

Thank you for this observation, this will be fixed.

**3  Minor Corrections and things to consider (not necessary changes)**

**3.1** Line 72 - grammar - consider switching "while occurring" with "that occur"

Thank you for the suggestion, this will be corrected.

**3.2** Line 205 - grammar - you are missing a word in "The weather stations measure hourly air temperature and relative humidity ___ sensors mounted at 2 m above ground level" some possibilities are rewording or filling ___ with (using/from...)

Thank you, we propose modifying the sentence as follows:

Lines 205-207: *The weather stations measure hourly air temperature (model CS109, Campbell Scientific) and relative humidity (model HMP155a, Campbell Scientific)* **with** *sensors mounted at 2 m above ground level.*

**3.3** Line 201 - it could be cleaner for the reader if you edited to make "from weather stations near the disdrometer stations" the last statement in the paragraph lines 201-209, moved it to the beginning of the next paragraph lines 210-219.  As is, it leaves the reader wondering How close is near? - However, not a major issue since the answer is found in the next paragraph.

This is a good point. We suggest removing the part that mentions the distance between sites, as it is unnecessary in the context of the paragraph, and that the following paragraph provides more details about the distance:

Lines 201-203: *The meteorological observations in this study* **come from** *weather stations, operated by Hydro-Quebec and SOPFEU, the province's wildfire prevention organisation.*

**3.4** Line 220 - Formatting - Table 1 seems to be the last two words of the previous paragraph rather than needing a new line.

Thank you, the formatting will be corrected.

3.5     Lines 419 - 423 as it relates to Figure 6d - Something to check.  It seems a bit odd, but possible that the mean precipitation rate 0.9mm is higher than the medians of 0.8mm, 0.7mm, and 0.6mm for mixed phase, liquid phase, and solid phase respectfully.  It can be a correct statement, but figure 6d makes this less likely given the very low numbers of heavy precipitation events depicted in the chart...  It's not highly important to the paper, but does look a bit off.

The values presented have been double checked and are as presented in the manuscript. This result is probably due to the significant weight of the lower precipitation values, which make the median lower than the mean, rather than higher precipitation values that result in a mean greater than the median.

3.6     Line 452 - grammar - "the most the mixed phase" - perhaps PGP_basic has the greatest overprediction in mixed phase (plenty of options, but right now the grammar is incorrect).

Thank you, it will be corrected.

3.7     Lines 566 - 568 - wording is a bit tricky, no issues with the beginning " The layer thickness is affected by environmental temperatures, as air density is inversely proportional to its temperature" however the end needs to indicate temperatures increasing or decreasing to finish the thought "which increases the distance between two pressure levels." A suggestion would be "... temperature. Therefore, as temperatures increase, the distance between two pressure levels also increases".

Thank you, we propose modifying the sentence to follow this suggestion:

Lines 566-568: *The layer thickness is affected by environmental temperatures, as the air density is inversely proportional to its temperature.* ***Therefore, as temperatures increase, the distance between two pressure levels also increases.***

3.8     Line 679-680 - grammar (missing word) - "options such ___ laser disdrometers", looks like it should be "such as".

Thank you, it will be corrected.

3.9     Line 766 - 767 - I would suggest consulting co-authors to make sure this is the final consensus on why "The longer time-step may lead to a lower critical threshold because the energy needed to melt the precipitation can be supplied over a longer period", I can't attach this thought with anything in the paper and would not personally agree with this statement.

This is a good point, and the reasoning will be reworded for clarity. The goal of this statement was to express $CT$ differences could be due to the validation data's timestep, the sentence will be reworded.

In the case of both Jennings et al. (2018) and Dai (2008), the validation data are at a 3-h time step and the $CT$ is lower than this study's solid $CT_S$. Additionally, Jennings et al. (2018) showed even greater differences, as they separated the data into relative humidity and surface pressure bins, whereas Dai (2008) lumped all overland data together. Following this result, one could argue that the $CT$ would decrease for longer timesteps, where mixed-phase precipitation due to a phase change is more likely to occur and requires overall colder temperatures for solid precipitation. Figure R1 illustrates the evolution of $CT_S$ from the dataset, which has been resampled to increasingly longer time steps. The longer the timestep, the colder $CT_S$ becomes. The $CT_S$ for humid conditions (i.e., greater than 90% relative humidity) are slightly colder and show a decrease similar to the full data curve.

[Figure]

Figure R1: $CT_S$ according to resampled data timestep, for the full dataset and high humidity data points (> 90% relative humidity).

As a matter of fact, the 3-hourly $CT_S$ decreases to 1.2°C and is in line with the value for all overland observations from Dai (2008). When accounting for humid conditions, $CT_S$ further decreases to 1.1°C. A noticeable difference remains however, especially compared to Jennings et al. (2018), even when accounting for the timestep and relative humidity range. The difference could come from the phase identification errors of both validation data sources, as it seems that the disdrometers used identify solid precipitation at warmer temperatures than the other studies mentioned.

We propose first refining slightly the comparison with the results in Behrangi et al. (2018) to show the difference between $CT_S$ and their $CT_a$ for humid conditions:

Lines 743-746: *One of the main conclusions of the study was that the wet-bulb temperature model is more robust than the dry-bulb temperature model because the $CT_a$ can vary significantly from site to site.* ***As such, the $CT_a$ for humid conditions would be approximately equal to the mean value minus the standard deviation, resulting in 1.18°C. The $CT_a$ for humid conditions is thus much closer to the $CT_S$ of 1.3°C in this study. Additionally, the*** *upper limit* ***of the $CT_a$ of*** *2.16°C in Behrangi et al. (2018)* *closely matches the $P_m$ of this study.*

Then we suggest a change at the end of the paragraph containing the problematic sentence that is the source of the reviewer's concern:

Lines 764-767: ***The greater difference between these $CT_a$ and $CT_S$ could be due to several reasons. First, the 3-hourly $CT_a$ should theoretically be lower than the hourly $CT_a$. As the timestep increases, the occurrence of mixed-phase precipitation increases due to the higher likelihood of a phase transition. Second, the different types of validation data could explain why $CT_a$ is generally lower than $CT_S$. Phase identification errors, particularly near the solid-liquid phase transition, could differ between direct observations and radar disdrometers.***

Finally, we propose a small addition to the following lines to reflect the previous modifications:

Lines 768-769: *Overall, the radar-based disdrometer measurements are similar to the findings of previous studies,* ***although with generally slightly warmer conditions of occurrence for solid precipitation. However,*** *more research is needed to properly quantify the uncertainties associated with this type of disdrometer.*

3.10   Line 792 - 793 - should double check this, might be 850-1000mb height difference according to figures 10 and 11 - "According to the input variable importance analysis, *atmospheric pressure* was the second most important hydrometeorological variable for phase classification" - It is the second greatest reanalysis variable (bright blue in figure 11) but the 4th most important variable if considering all data in Figure 11.  This statement could be correct depending on the intended meaning of "hydrometeorological variable".

This is indeed ambiguous. We propose adding "near-surface" to the sentence for precision:

Lines 792-793: *According to the input variable importance analysis, atmospheric pressure was the second most important **near-surface** variable for phase classification.*

3.11   Lines 811 - 814 (Appendix A) - Longitude and Latitude, some values are given to 5 decimal points and others to 6, usually these values all have similar accuracy.  I would suggest either rounding to 5 decimals, or if it is dropping a sixth decimal if = to 0, reformatting to show the 0 to show all coordinates having accuracy to 6 decimal points.

This is a good point, a zero will be added in the instances where there are only 5 decimal points.

**4  References to be added to the manuscript**

WMO: 6.1.2 Units and scales, in: Measurement of Meteorological Variables, 2023 edition ed., edited by: WMO, Guide to instruments and methods of observation, Volume 1, WMO, Geneva, 574, 2018.

**5  References used in this document**

Behrangi, A., Yin, X., Rajagopal, S., Stampoulis, D., and Ye, H.: On distinguishing snowfall from rainfall using near-surface atmospheric information: Comparative analysis, uncertainties and hydrologic importance, Quarterly Journal of the Royal Meteorological Society, 144, 89-102, https://doi.org/10.1002/qj.3240, 2018.

Dai, A.: Temperature and pressure dependence of the rain-snow phase transition over land and ocean, Geophysical Research Letters, 35, https://doi.org/10.1029/2008GL033295, 2008.

Jennings, K. S., Winchell, T. S., Livneh, B., and Molotch, N. P.: Spatial variation of the rain-snow temperature threshold across the Northern Hemisphere, Nat Commun, 9, 1148, https://doi.org/10.1038/s41467-018-03629-7, 2018.

---

## Author Response (AR2)

**HESS-2024-78 Leveraging a Disdrometer Network to Develop a Probabilistic Precipitation Phase Model in Eastern Canada**

**1 Response to anonymous referee 1**

The authors have addressed my review comments. I noted a few typographical corrections while reading this revised manuscript, but I do not need to see it again before publication.

Thank you for your final feedback, it is greatly appreciated. Please note that the line numbers in this section refer to the lines in the new version of the manuscript submitted with this response document.

**1.1 Line 90: "droplets" refers to liquid precipitation only**

Thank you, "droplets" was replaced by "precipitation particles".

**1.2 Line 92: "the hydrometeors exchange latent and sensible heat with their surroundings"**

Thank you, the sentence was fixed.

**1.3 Line 130: "hourly or shorter timesteps"?**

Thank you for pointing out this mistake, it was fixed.

**1.4 Line 257: "noise and diurnal oscillations in the precipitation data"**

This was fixed, thank you.

**1.5 Line 559: "The thinner air in low-pressure environments"**

This was fixed, thank you.

**2  Additional changes made to the manuscript**

Please note that the line numbers in this section refer to the lines in the new version of the manuscript submitted with this response document. Additions to the manuscript are shown in bold.

**2.1  Typographical corrections**

Line 153: "observations" to "observation"

Line 196, 204, 262 and 710: "identifications" to "identification"

Line 228: "atmosphere" to "atmospheric levels"

Line 263: removed the second "respectively" at the end of the sentence

Line 280: "results" to "result"

Line 466: "best performing" to "best-performing"

Line 534: "increases" to "increase"

Line 724: "large amount of stations" to "large number of stations"

Line 1075: "station" to "stations"

Line 1080: "modeled" to "modelled"

Line 1081: added "(" before figure subsections in caption for consistency

**2.2  Technical corrections**

**2.2.1  Line 176: The network lies between latitudes 47.23 and 52.13 °N, and longitudes 63.17 to 75.29° W, spanning an area of roughly 1,138,000 km$^2$.**

The projected area was corrected to be more accurate at the latitudes of the study:

Lines 175-176: *The network lies between latitudes 47.23 and 52.13 °N, and longitudes 63.17 to 75.29° W, spanning an area of roughly **532,594** km$^2$.*

**2.2.2  Lines 788-789**

A sentence to introduce Table A1 was added:

Lines 788-789: **Table A1 shows the details for the study sites used from the Hydro-Québec observational network.**